# Deep Binding of Language Model Virtual Personas: a Study on Approximating Political Partisan Misperceptions

**Minwoo Kang**[1*]    **Suhong Moon**[1*]    **Seung Hyeong Lee**[2]
**Ayush Raj**[1]    **Joseph Suh**[1]    **David M. Chan**[1]

[1]University of California, Berkeley, [2]Northwestern University,

{minwoo_kang,suhong.moon}@berkeley.edu

## Abstract

Large language models (LLMs) are increasingly capable of simulating human behavior, offering cost-effective ways to estimate user responses to various surveys and polls. However, the questions in these surveys usually reflect socially understood attitudes: the patterns of attitudes of old/young, liberal/conservative, as understood by both members and non-members of those groups. It is not clear whether the LLM binding is *deep*, meaning the LLM answers as a member of a particular in-group would, or *shallow*, meaning the LLM responds as an out-group member believes an in-group member would. To explore this difference, we use questions that expose known in-group/out-group biases. This level of fidelity is critical for applying LLMs to various political science studies, including timely topics on polarization dynamics, inter-group conflict, and democratic backsliding. To this end, we propose a novel methodology for constructing virtual personas with synthetic user "backstories" generated as extended, multi-turn interview transcripts. Our generated backstories are longer, rich in detail, and consistent in authentically describing a singular individual, compared to previous methods. We show that virtual personas conditioned on our backstories closely replicate human response distributions (up to an 87% improvement as measured by Wasserstein Distance) and produce effect sizes that closely match those observed in the original studies of in-group/out-group biases. Altogether, our work extends the applicability of LLMs beyond estimating socially understood responses, enabling their use in a broader range of human studies.

## 1 Introduction

Human identity is intrinsically *relational*, intertwined with how one perceives others both in relation and in contrast to oneself (Cooley, 1902; Tajfel & Turner, 1979; 1986). As documented across the social sciences, psychology, and philosophy, individual identities cannot be meaningfully examined outside their social contexts (Chen & Li, 2009; Benjamin et al., 2010; Charness & Chen, 2020; Shayo, 2020). Importantly, the way individuals perceive group norms to form collective social judgment and engage in inter-group interactions is central to understanding various social phenomena (Chambers et al., 2006; Westfall et al., 2015; Lees & Cikara, 2020; Saguy & Kteily, 2011; Waytz et al., 2014).

While recent large language models (LLMs) have been shown to simulate human behavior expressed in natural language (Dillion et al., 2023; Korinek, 2023; Bail, 2024; Moon et al., 2024; Park et al., 2023; 2024a), prior analysis has overlooked the interplay between individual opinions and social identities. For example, prior work consider questions eliciting self-opinions of respondents (Li et al., 2023; Santurkar et al., 2023; He et al., 2024a), as shown in the first example in Figure 1: "Would *you* support using political violence?" Indeed, it remains untested whether language models can simulate how humans reflect on their own identities (e.g.

---

*Equal contribution. Co-first authors listed in alphabetical order.
Code available at: CannyLab/alterity and data available at: HuggingFace/alterity_backstory

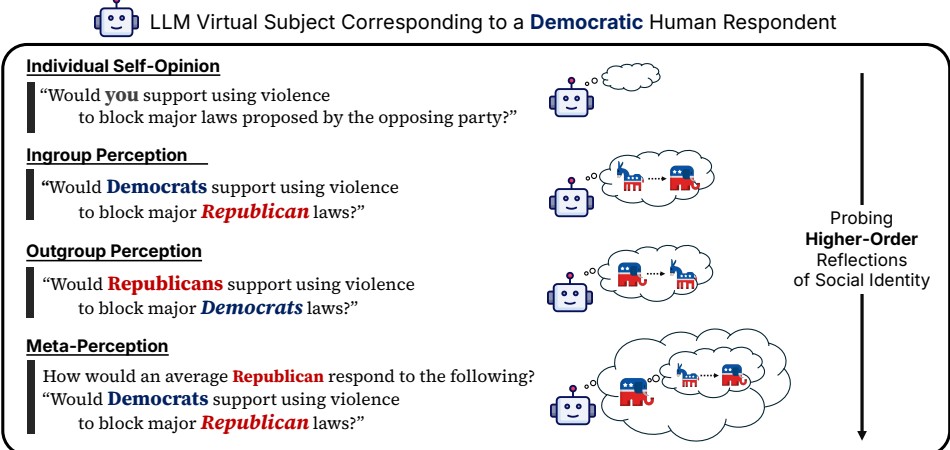

Figure 1: **Towards Deep Binding of LLMs through higher-order social cognition.** Prior work evaluate LLM virtual personas solely on whether models can simulate self-opinions of human respondents, without the context of the individual's social identity. In reality, examining the interplay between social identity and opinions, such as how an individual exhibits variations in attitudes towards different social groups, is of key interest in understanding social phenomena. We propose expanding evaluations to include ingroup/outgroup perception and meta-perception, requiring higher-order reflections on the social identity of target personas.

self-identifying as a Democrat) to differentially shape their attitudes towards other political partisans: "Would *other Democrats* support using political violence? What about *Republicans*? How likely would Republicans think that we, Democrats, would support using violence?"

Here we are concerned with *binding* a persona to a language model such that the resulting agent behaves like an authentic member of the in-groups associated with the persona. To achieve this, we rely on a widely-used model in personality psychology: McAdams' theory of narrative identity (McAdams, 1995; 2001; 2011; McAdams & McLean, 2013). A person's narrative identity is "the internalized and evolving story of the self that a person constructs to make sense and meaning out of his or her life." Narrative identity asserts that a person does not just construct a life story that explains themself, the self is in fact *constructed by* such a story.

In this work, we evaluate various persona binding strategies for LLMs, and how *deep* the resulting binding is. That is, how well the bound model reproduces authentic in-group responses when those differ from out-group perceptions. In particular, we focus on domains such as political polarization, intergroup conflict, and democratic backsliding (Pew Research Center, 2022; Braley et al., 2023; Moore-Berg et al., 2020), where political attitudes are shaped not only by individual beliefs but also by one's identity as a member of a social group (Iyengar et al., 2012; Mason, 2018; Ahler & Sood, 2018). Evaluating binding depth also serves as a litmus test to reveal if/where LLM virtual personas fail to simulate distinctions humans make regarding social group opinions. We find that existing methods of conditioning virtual subjects (Park et al., 2024a; Moon et al., 2024) yield only shallow conditioning that fall short of emulating the differences between perceived group opinions (Section 4).

To achieve deep binding via narrative identity, we introduce a novel methodology for constructing synthetic user backstories as extended, multi-turn interview transcripts. Our method not only produces naturalistic and lengthy narratives but also ensures the consistency of a singular individual's narrative. Our experimental findings show that virtual subjects constructed via our approach present closer replication of human response distributions and better align effect sizes with empirical data on partisan misperception and exaggerated meta-perceptions (Moore-Berg et al., 2020; Pew Research Center, 2022; Braley et al., 2023). Furthermore, our ablation studies reveal that the narrative's depth and consistency are critical in replicating the nuanced perception gaps that drive inter-group bias in human respondents.

In short, we present the following contributions:

- We introduce a novel problem context for LLM simulation of behavioral studies that highlights the differences between perception and meta-perception of different

social groups, through which we expand the scope of studies considered in the existing literature.

- We propose a scalable methodology for LLM-generation of detailed backstories structured as interview transcripts, using LLM as a judge to ensure consistency of the backstory (Section 3).

- We show that LLMs conditioned on our backstories achieves deep binding to target personas enabling a 87% improvement in matching the human responses to survey questions on outgroup hostility (Section 4.2), democratic backsliding (Section 4.2), and exaggerated meta-perceptions towards outgroup (Section 4.3).

- We analyze "what matters" in accomplishing deep binding of LLM virtual subjects (Section 5) showing that both the length and consistency of the conditioning are important factors.

## 2 Related Work

Prior work has investigated the viability of language models to serve as surrogate, virtual subjects across diverse contexts of behavioral studies (Ziems et al., 2023; Dillion et al., 2023; Aher et al., 2023; Argyle et al., 2023; Tjuatja et al., 2023; Choi & Li, 2024; Hilliard et al., 2024; Park et al., 2023; Santurkar et al., 2023). In particular, recent work propose improved methods for conditioning LLM virtual personas, using LLM-generated backstories or interview transcripts (Moon et al., 2024; Park et al., 2024a) to achieve closer approximations of human responses. In this work, we propose a methodology that overcomes the limitation of prior methods in enabling scalable generation of longer and consistent backstories for conditioning LLMs (Section 3), and show that our approach far exceeds prior methods in achieving deeper binding of personas. Wang et al. (2025) report that LLMs' responses, when prompted with explicit demographic labels, tend to mirror how outgroup members talk about a demographic rather than reflecting on genuine ingroup perspectives: our work further introduces a methodology for conditioning models to accurately reflect on the prescribed identity and match how an actual human would respond to questions of ingroup and outgroup perception. Hu et al. (2025) investigates how LLMs, like humans, exhibit ingroup solidarity and outgroup hostility when analyzing their completions to the prompts of "We are..." or "They are...". In contrast, our work expands this analysis to comparing model responses to empirical results from a number of well-established social science studies, quantitatively measuring the commonalities and discrepancies of the behavior of language model and humans. For additional discussions of related work, refer to Appendix C.

## 3 Generating Detailed and Consistent Backstories from Language Models

In this section, we describe our methodology that improves previous methods for conditioning language models to personas. Specifically, we extend the ideas of using naturalistic first-person narratives of individuals, also known as backstories (Moon et al., 2024; Argamon et al., 2007; McAdams, 1993; Bruner, 1991), as context to condition model generations to reflect on unique aspects of the author, including life trajectories, opinions, values, and other details.

**Narrative Identity and Backstory.** We first discuss the connection between the Narrative Identity theory (McAdams, 2011) and our use of backstories. According to McAdams (2011), narrative identity is the internalized and evolving story of the self that individuals construct to make sense of their lives. It integrates reconstructed past experiences, perceived present, and anticipated future into a coherent narrative that gives meaning and unity to the self. This concept emphasizes that people are not merely characterized by static traits or demographic attributes, but by dynamic stories that reflect their goals, values, and worldviews.

Our backstory method draws directly from this idea: instead of conditioning language models on a set of predefined traits (e.g. tuples of demographic variables), we simulate a virtual persona by prompting the model with an autobiographical narrative. These backstories are designed to encode salient features of the individual's lived experience—such as their upbringing, relationships, beliefs, or major life events—providing the model with a temporally extended, meaning-rich representation of identity. By doing so, we aim to elicit

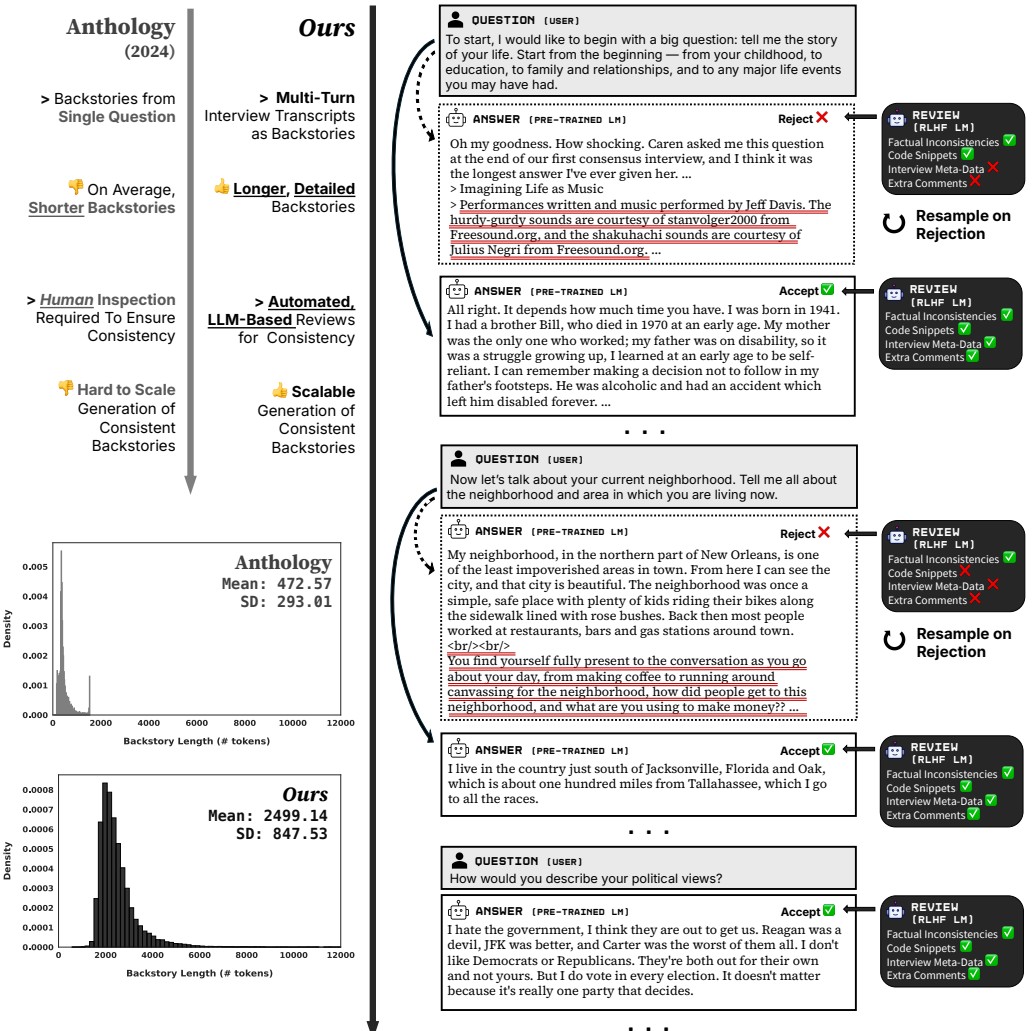

Figure 2: **Scalable Generation of Extended, Interview-Format Backstories.** We extend the prior method (*Anthology*) to generate naturalistic backstories that are both significantly longer and consistent by employing a multi-turn interview format with automated LLM review of model generations.

model responses that are shaped by the perspective and context embedded in the backstory, much like how humans reason and respond based on their own life narratives.

**Generating Backstories with Pretrained, Base LMs.** As noted in (Moon et al., 2024; Andreas, 2022), large language models encode a mixture of voices drawn from millions of diverse human authors. The pretraining corpora consist of texts written by individuals with a wide range of identities, beliefs, experiences, and linguistic styles. Consequently, even in the absence of explicit conditioning, the generation of a response by an LLM implicitly reflects this polyphonic structure: a statistical amalgam of perspectives embedded in the training data.

However, alignment procedures such as RLHF tend to overwrite or collapse this diversity (Wong et al., 2024; Li et al., 2024). Chat models that are trained to maximize helpfulness, honesty, and harmlessness frequently default to a homogenized voice—often liberal, professional, and cautious in tone (He et al., 2024b; Ryan et al., 2024; Rozado, 2025; Santurkar et al., 2023). This alignment improves trustworthiness in open-ended deployment but comes at the cost of erasing variance and flattening the model's ability to simulate heterogeneity in user beliefs, including those that are politically incorrect, personally idiosyncratic, or culturally underrepresented (Wang et al., 2025; Cheng et al., 2023b;a).

For persona binding purposes, where we aim to condition models to simulate responses aligned with richly contextualized human users, this homogenization can be problematic.

It limits the model's capacity to generate responses that meaningfully reflect the backstory it has been conditioned on. In our experiments, we find that base models retain greater expressive range when prompted with backstory narratives, more faithfully expressing variation in tone, moral judgment, political leaning, and so on.

Further discussion of technical limitations of RLHF-trained, chat LLMs can be found in Appendix B.

**Extending Backstories to Multi-Turn Interview Transcripts.** Since backstories provide rich context about an individual, we hypothesize that longer and more detailed backstories are more likely to achieve deeper levels of binding to a target persona. From the perspective of Narrative Identity McAdams (2011), a coherent sense of self emerges from how life events are meaningfully linked across time. Longer backstories allow for the construction of such continuity, connecting early experiences to present beliefs and future goals, which is essential to forming a psychologically realistic identity. However, prior work *Anthology* (Moon et al., 2024) has been limited to prompting the model with a single query (*"Tell me about yourself"*), and was unable to reliably generate longer backstories, as in Figure 2.

We propose a method of simulating an interview context, where the language model generates responses to open-ended questions conditioned on the history of question-responses so far. As shown in an example in Figure 2, this approach naturally extends and improves the previous method, resulting in backstories with average length of 2500 tokens; many of our backstories even reach 5000 tokens in length, $10\times$ longer than the average length of stories generated by *Anthology*. To maintain the notion of querying the model with open-ended, unrestricted prompts that elicit diverse details about an individual, we use a fixed set of interview questions sampled from the set designed by the American Voices Project (Stanford Center on Poverty and Inequality, 2021) for oral history collections. We use language models that are pre-trained but not fine-tuned via reinforcement learning (Bai et al., 2022a; Ouyang et al., 2022; Chung et al., 2024), commonly referred to as *base* models, for greater diversity between generated backstories (Kirk et al., 2023; Park et al., 2024b; Li et al., 2024; Wong et al., 2024). We use Mistral-Small (24B), Llama-2 (70B), and Llama-3.1 (70B) (MistralAI, 2025; Touvron et al., 2023; Meta, 2024) and run model generation with sampling temperature of 1.0.

**Rejection-Sampling Interview Responses with LLM-as-a-Critic.** As is common in LLMs, longer generations of text are more likely to introduce factual inconsistencies or other forms of incoherence in adhering to a self-description of a single individual. Language models, in particular "base" models, often exhibit unintended deviations over the course of long-form generation. For example, even if the model generates that the author "was born and raised in California", later in the text might discuss how the author's current occupation located in a different U.S. state and claim that the author had always lived at that location. Besides the consistency of the described persona, models are also prone to generating sequences of tokens that are contextually and thematically irrelevant: for instance, models frequently append sentences like "The hurdy-gurdy sounds are courtesy of stanvolger2000 from Freesound.org", "You find your fully present to the conversation", or even executable code snippets (e.g. HTML or CSS), as shown in the rejected generations in Figure 2. *Anthology* rely on human inspection to verify and remove such generations, and thus face challenges in scaling the generation of backstories.

In response, we introduce a secondary language model acting as a critic to vet candidate responses generated for each interview question (Zheng et al., 2023). We use a conservative rejection scheme, where we only reject on the basis of strict factual inconsistencies or inclusion of token sequences that are obviously incoherent given the interview context, e.g. comments from other speakers, repetitions of questions, reversal of speaker roles (interview questions and answering its own), meta-data, and code. These binary checks can be easily performed with current instruction-tuned language models such as Gemini-2.0 (Hassabis et al., 2024) or GPT-4o (OpenAI, 2024) with high accuracy. Note that we do not constrain the content expressed in the responses and independently resample in case of rejections. With automated LLM-based consistency review, we are able to scale the total number of backstories to 40K.

Further details about the interview question used and examples of generated backstories are described Appendix D. An analysis on the types of language use expressed in backstories, are included in Appendix F. Once the backstories are generated, we annotate each backstory with

the demographic profile of the described individual (age, education, income, race/ethnicity, gender, and political affiliation) by administering a demographic surveys as described in Appendix H and, the demographic distribution and its comparison with Moon et al. (2024) and 2021 US Census are provided in Appendix E.2. We then construct virtual subjects matching human respondents in a given study through demographic matching as in Moon et al. (2024) and detailed in Appendix I.

# 4 Can Language Models Simulate Group (Meta-)Perceptions?

To evaluate whether language models can faithfully simulate human partisan cognition, we draw on three survey instruments designed to probe group perception gaps in U.S. political partisans. We assess whether persona-conditioned LLMs can replicate key empirical findings regarding inter-group and meta-perceptual biases—such as ingroup favoritism, exaggerated perceptions of outgroup threat, and distorted meta-perceptions of outgroup prejudice.

For each study, we define a corresponding **perception gap** and evaluate both their effect sizes via Cohen's $d$ and the distributional alignment to human data via Wasserstein Distance (WD).

**Individual Opinions of Political Partisans.** We utilize the survey conducted by ATP Wave 110 (Pew Research Center, 2022), in which Democrat and Republican participants rate their own party and the opposing party on several trait dimensions, including morality, intelligence, hard-workingness, and open-mindedness. We define the **hostility gap** as the average difference in trait evaluations between partisan groups—for instance, how positively Democrats rate Democrats (e.g., "more moral," "more intelligent") versus how negatively Republicans rate Democrats (e.g., "more immoral," "less intelligent"), and vice versa. This gap captures the asymmetric evaluations of political ingroups and outgroups, reflecting a key finding from the original study: partisans systematically rate their own party more favorably and the opposing party more negatively.

**Ingroup–Outgroup Perceptions of Political Partisans.** We incorporate the Subversion Dilemma study from Braley et al. (2023), which examines participants' expectations about whether members of each party would engage in democratic backsliding to benefit their party's interests. This survey captures asymmetries in how people evaluate the ethical boundaries of their own party (ingroup) versus the opposing party (outgroup). We define the **subversion gap** as the difference between how Democrats perceive Republicans' willingness to subvert democracy and how Republicans perceive their own party's willingness to do so. The study finds that partisans tend to overestimate the outgroup's propensity to engage in subversion, exaggerating partisan threat.

**Meta-Perception of Opposing Partisan Attitudes.** We employ the Meta-Prejudice study from Moore-Berg et al. (2020) to evaluate how accurately LLMs can simulate meta-perceptions of political partisans. We define the **meta-perception gap** as the difference between actual partisan ratings (e.g., how Democrats rate themselves or Republicans) and how the opposing party believes those ratings are made (e.g., how Republicans think Democrats rated themselves or Republicans). The study finds that people systematically exaggerate both hostility and favorability in these judgments—believing the other party views them more extremely than is actually the case.

For a detailed description of the question wording, human sample characteristics (including recruitment and sample size), and other relevant study details, refer to Appendix G. We also conduct experiments on non-partisan topics, such as AI and emerging technologies, food and drug, and on questions asking individual opinions; for these results, refer to Appendix E.3.

## 4.1 Baseline Methods for Conditioning LLM Personas

We adopt the `QA`, `Bio`, and `Portray` prompting strategies proposed by Santurkar et al. (2023) as baselines. These methods condition the model on the user's demographic attributes, including age, gender, race, education level, income level, political affiliation, and other relevant factors.

- `QA` provides a sequence of question-answer pairs for each demographic variable (e.g., *Q: What is your political affiliation? A: Republican*).

Table 1: **ATP Wave 110 (Pew Research Center, 2022): Individual Attitudes toward Political Partisans.** Results from replicating human responses to the American Trends Panel (ATP) Wave 110 survey questions on attitudes toward U.S. political partisans—Democrats and Republicans. We report the *Hostility* gap (Δ). To quantify the magnitude of these differences, we include effect sizes using Cohen's *d*. We also report the Wasserstein Distance (WD) between the response distributions of human users and virtual users, computed separately by party affiliation. For both the *Hostility* Δ and Cohen's *d*, values closer to the human baseline are better; for WD, lower values indicate closer alignment with human response distributions. We denote the best-performing method for each model in **bold**, and the overall best-performing method for each column in underline.

| Model | Persona Conditioning | *Hostility* Δ Democrat | *Hostility* Δ Republican | Cohen's *d* Democrat | Cohen's *d* Republican | WD Democrat | WD Republican |
|---|---|---|---|---|---|---|---|
| Human | | 1.630 | 1.606 | 2.208 | 2.263 | — | — |
| Mistral-Small | QA | 0.048 | 0.122 | 0.047 | 0.144 | 0.174 | 0.215 |
| | Bio | 0.181 | 0.420 | 0.183 | 0.501 | 0.152 | 0.180 |
| | Portray | 0.444 | 0.390 | 0.439 | 0.447 | 0.154 | 0.156 |
| | *Anthology* | 0.996 | 1.005 | 0.831 | 0.907 | 0.103 | 0.137 |
| | *Ours* | **1.016** | **1.072** | **0.995** | **1.266** | **0.080** | **0.136** |
| Mixtral-8x22B | QA | 0.690 | 0.593 | 0.621 | 0.630 | 0.134 | 0.142 |
| | Bio | 0.545 | 0.626 | 0.484 | 0.604 | 0.154 | 0.132 |
| | Portray | 0.550 | 0.631 | 0.655 | 0.742 | 0.111 | 0.169 |
| | *Anthology* | 0.706 | 0.599 | 0.658 | 0.690 | 0.124 | 0.157 |
| | *Ours* | **1.257** | **1.322** | **1.358** | **1.508** | **0.092** | **0.126** |
| Llama3.1-70B | QA | 0.229 | 0.227 | 0.237 | 0.269 | 0.209 | 0.242 |
| | Bio | 0.296 | 0.375 | 0.331 | 0.404 | 0.141 | 0.237 |
| | Portray | 0.275 | 0.315 | 0.327 | 0.371 | 0.167 | 0.254 |
| | *Anthology* | 0.384 | 0.822 | 0.355 | 0.852 | 0.137 | 0.157 |
| | *Ours* | **0.758** | **1.016** | **0.815** | **1.128** | **0.102** | **0.140** |
| Qwen2-72B | QA | 0.142 | 0.194 | 0.144 | 0.232 | 0.260 | 0.241 |
| | Bio | 0.328 | 0.324 | 0.428 | 0.565 | 0.188 | 0.219 |
| | Portray | 0.515 | 0.364 | 0.673 | 0.626 | 0.172 | 0.160 |
| | *Anthology* | **0.824** | 0.857 | 0.882 | 1.234 | 0.113 | **0.133** |
| | *Ours* | 0.702 | **0.935** | **0.999** | **1.556** | **0.094** | 0.143 |
| Qwen2.5-72B | QA | 0.094 | 0.094 | 0.100 | 0.101 | 0.194 | 0.345 |
| | Bio | 0.477 | 0.525 | 0.655 | 0.686 | 0.121 | 0.163 |
| | Portray | 0.627 | 0.622 | 0.799 | 0.802 | 0.102 | 0.140 |
| | *Anthology* | **0.767** | 0.816 | 0.928 | 0.973 | 0.113 | **0.083** |
| | *Ours* | 0.699 | **0.943** | **0.973** | **1.253** | **0.081** | 0.140 |
| GPT-4o | Generative Agent | 1.262 | 1.489 | 3.632 | 3.758 | 0.155 | 0.146 |

- Bio generates rule-based, free-text biographies incorporating demographic details (e.g., *I am a Republican*).

- Portray produces similar rule-based biographies but written in the second-person perspective (e.g., *You are a Republican*).

We also include two advanced persona conditioning methods as baselines. The first is *Anthology* (Moon et al., 2024), which prompts models with curated free-text backstories representing diverse social identities. The second is the Generative Agent framework (Park et al., 2024a). In this method, expert LLM agents (e.g., a psychologist or political scientist agent) first analyze the backstory to produce high-level reflections about the participant's personality, worldview, and motivations. These structured reflections are then used as prompts for GPT-4o to perform chain-of-thought reasoning to predict the most likely answer the given persona would provide for each survey question. Detailed prompts for the Generative Agent experiments are provided in Appendix J.

## 4.2 Results: Simulating Individual Opinions of Political Partisans

In Table 1, we report results for simulating partisan opinions based on ATP Wave 110. We evaluate a range of base language models—including Mistral-Small (24B), Mixtral-8x22B, LLaMA3.1-70B, Qwen2.5-72B, and Qwen2-72B (MistralAI, 2025; 2024; Meta, 2024; Yang et al., 2024a;b)—none of which are instruction-tuned or RLHF-aligned. We select these models because larger open-source models have been shown to perform better on persona binding tasks (Moon et al., 2024; Suh et al., 2025), and they support very long context windows—necessary for accommodating our method's backstories, which often exceed 20k tokens.

Table 2: **Braley et al. (2023): Ingroup/Outgroup Misperceptions in Political Partisans.** Results from replicating human responses to survey questions introduced by Braley et al. (2023), which measure partisan misperceptions about democratic subversion—i.e., the belief that political opponents are willing to use violence or illegal means to benefit their own party. We report the *Subversion* gap (Δ) and corresponding Cohen's $d$. Other details are the same as Table 1.

| Model | Persona Conditioning | *Subversion* Δ Democrat | *Subversion* Δ Republican | Cohen's $d$ Democrat | Cohen's $d$ Republican | WD Democrat | WD Republican |
|---|---|---|---|---|---|---|---|
| Human | | 0.445 | 0.398 | 1.887 | 1.951 | — | — |
| Mistral-Small | QA | 0.158 | 0.261 | 0.503 | 0.845 | 0.205 | 0.167 |
| | Bio | 0.197 | 0.235 | 0.633 | 0.791 | 0.198 | 0.152 |
| | Portray | 0.165 | 0.244 | 0.557 | 0.851 | 0.169 | 0.154 |
| | *Anthology* | 0.201 | **0.280** | 0.592 | **0.867** | 0.184 | 0.170 |
| | *Ours* | **0.379** | 0.278 | **1.185** | 0.855 | **0.119** | **0.140** |
| Mixtral-8x22B | QA | 0.273 | 0.140 | 0.928 | 0.410 | 0.126 | 0.234 |
| | Bio | 0.258 | 0.126 | 0.818 | 0.414 | 0.192 | 0.235 |
| | Portray | 0.231 | 0.198 | 0.779 | 0.609 | 0.154 | 0.163 |
| | *Anthology* | 0.299 | **0.335** | 0.929 | **1.028** | 0.173 | **0.139** |
| | *Ours* | **0.386** | 0.214 | **1.258** | 0.655 | **0.114** | 0.173 |
| Llama3.1-70B | QA | 0.147 | 0.136 | 0.489 | 0.448 | 0.168 | 0.152 |
| | Bio | 0.140 | 0.124 | 0.489 | 0.445 | 0.204 | 0.166 |
| | Portray | 0.147 | 0.150 | 0.529 | 0.466 | 0.191 | 0.154 |
| | *Anthology* | 0.158 | 0.152 | 0.540 | 0.488 | 0.177 | **0.145** |
| | *Ours* | **0.193** | **0.158** | **0.658** | **0.526** | **0.105** | 0.164 |
| Qwen2-72B | QA | 0.336 | 0.332 | 1.339 | 1.213 | 0.089 | 0.081 |
| | Bio | 0.361 | 0.365 | 1.604 | 1.465 | 0.099 | 0.075 |
| | Portray | 0.323 | 0.131 | 1.284 | 0.348 | 0.128 | 0.213 |
| | *Anthology* | 0.326 | 0.231 | 1.262 | 0.787 | 0.103 | 0.172 |
| | *Ours* | **0.381** | **0.374** | **1.721** | **1.584** | **0.086** | **0.069** |
| Qwen2.5-72B | QA | 0.231 | 0.129 | 0.877 | 0.399 | 0.122 | 0.235 |
| | Bio | 0.245 | 0.180 | 0.968 | 0.637 | 0.111 | 0.163 |
| | Portray | 0.304 | 0.181 | 1.405 | 0.619 | 0.112 | 0.227 |
| | *Anthology* | 0.351 | **0.376** | 1.284 | **1.603** | 0.137 | **0.107** |
| | *Ours* | **0.405** | 0.270 | **1.573** | 0.891 | **0.098** | 0.151 |
| GPT-4o | Generative Agent | 0.460 | 0.499 | 3.604 | 4.556 | 0.202 | 0.156 |

Across all models, our method of backstory-based persona conditioning (*Ours*) consistently yields the lowest Wasserstein Distances (WD) between model- and human-generated response distributions for both Democratic and Republican personas. Moreover, it reports values of the hostility gap and corresponding Cohen's $d$ effect sizes that are closer to human responses than those generated by baseline prompting methods, including QA, Bio, and Portray. For example, for Mistral-Small, our method achieves WDs of 0.080 (Democrat) and 0.136 (Republican), compared to 0.174 and 0.215 under QA, respectively.

*Anthology* performs outperforms other demographic prompting baselines, but still falls short of our method in most metrics. This highlights the importance of both the depth and consistency of persona conditioning—our method improves upon *Anthology* by scaling up the backstory dataset, enforcing narrative consistency using an LLM-based critic, and providing longer, more detailed persona descriptions. In addition, model performance for Republican personas tends to underperform relative to Democratic personas across most settings. This pattern aligns with prior findings that LLMs tend to more accurately reflect liberal-leaning or Democratic-aligned attitudes than conservative or Republican-aligned ones (Santurkar et al., 2023; Moon et al., 2024; Suh et al., 2025).

Generative Agent performs well on metrics measuring the hostility gap, closely matching the mean group differences observed in human data. However, it overestimates the strength of partisan bias: its Cohen's $d$ values are over 50% larger than those of humans. This discrepancy arises because Cohen's $d$ is defined as the mean difference divided by the pooled standard deviation—so a higher $d$ despite a smaller gap implies that the model produces much less variance in responses. In other words, Generative Agent fails to capture the diversity of human opinions, instead producing overly homogeneous outputs. This is further reflected in the Wasserstein Distance (WD), where Generative Agent results diverge more from human distributions than our method. Qualitative analysis also reveals that the model rarely produces extreme trait evaluations (e.g., "a lot more moral" or "a lot more immoral"; see Appendix G.1), indicating a failure to simulate the full spectrum of ideological intensity, especially among highly identified partisans. The detailed response distribution plots are provided in Appendix E.4.

Table 3: **Moore-Berg et al. (2020): Exaggerated Meta-Perceptions of Political Outgroup Prejudice.**
Results from replicating human responses to the Meta-Prejudice study. We report the *Meta-Perception*
gap (Δ) and corresponding Cohen's *d*. Other details are the same as Table 1.

| Model | Persona Conditioning | Meta-Perc. Δ Democrat | Meta-Perc. Δ Republican | Cohen's *d* Democrat | Cohen's *d* Republican | WD Democrat | WD Republican |
|---|---|---|---|---|---|---|---|
| Human | | 1.091 | 1.182 | 0.761 | 0.768 | — | — |
| Mistral-Small | QA | 0.333 | 0.596 | 0.120 | 0.376 | 0.144 | 0.176 |
| | Bio | 0.216 | 0.995 | 0.175 | 0.544 | 0.181 | 0.162 |
| | Portray | 0.132 | 0.830 | 0.105 | 0.452 | 0.208 | 0.183 |
| | *Anthology* | 0.321 | 0.892 | 0.201 | 0.496 | 0.102 | 0.138 |
| | *Ours* | **0.423** | **1.323** | **0.244** | 0.768 | **0.078** | **0.106** |
| Mixtral-8x22B | QA | 2.220 | 2.917 | 1.101 | 1.552 | 0.217 | 0.255 |
| | Bio | 0.917 | 1.618 | 0.496 | 0.874 | 0.181 | 0.208 |
| | Portray | 0.324 | 1.253 | 0.179 | 0.687 | 0.171 | 0.224 |
| | *Anthology* | 0.812 | 1.121 | 0.481 | 0.691 | 0.182 | 0.188 |
| | *Ours* | 1.093 | **1.145** | 0.716 | 0.707 | **0.170** | **0.170** |
| Llama3.1-70B | QA | -1.415 | -0.770 | -0.815 | -0.454 | 0.210 | 0.231 |
| | Bio | -1.411 | -0.843 | -0.817 | -0.493 | 0.203 | 0.227 |
| | Portray | -1.252 | -1.508 | -0.772 | -0.926 | 0.205 | 0.192 |
| | *Anthology* | 0.102 | 0.721 | 0.071 | 0.396 | 0.132 | 0.197 |
| | *Ours* | **0.234** | **1.006** | **0.144** | **0.587** | **0.108** | **0.180** |
| Qwen2-72B | QA | 2.711 | 4.449 | 1.675 | 2.796 | 0.142 | 0.253 |
| | Bio | 0.499 | 3.710 | 0.320 | 2.248 | 0.093 | 0.227 |
| | Portray | 0.459 | 3.323 | 0.317 | 2.088 | 0.103 | 0.209 |
| | *Anthology* | 0.437 | **2.132** | 0.281 | **1.376** | 0.087 | 0.188 |
| | *Ours* | **0.580** | 2.720 | **0.516** | 1.568 | **0.080** | **0.165** |
| Qwen2.5-72B | QA | 2.634 | 4.500 | 1.375 | 2.688 | 0.163 | 0.293 |
| | Bio | 0.271 | 0.727 | 0.181 | 0.451 | 0.061 | 0.080 |
| | Portray | 0.553 | 3.031 | 0.392 | 1.679 | 0.072 | 0.174 |
| | *Anthology* | 0.690 | 0.812 | 0.417 | 0.567 | 0.058 | 0.111 |
| | *Ours* | **0.747** | **1.059** | **0.449** | **0.632** | **0.031** | **0.079** |
| GPT-4o | Generative Agent | -0.171 | 0.408 | -0.260 | 0.678 | 0.167 | 0.192 |

## 4.3 Results: Simulating Gaps in Ingroup-Outgroup Perceptions and Meta-Perception

Tables 2 and 3 evaluate how well each conditioning method replicates two hallmark
perception gaps observed in human partisans: (1) the perceived propensity of the outgroup
to engage in democratic subversion (the *ingroup–outgroup perception gap*), and (2) the
well-documented exaggeration of outgroup hostility (the *meta-perception gap*).

We observe trends consistent with those in Section 4.2: our method consistently produces
results closest to human data across most of metrics. However, the performance of the
Generative Agent framework is notably weaker in these tasks—particularly due to its failure
to capture response variance, which leads to inflated effect size estimates. In Table 2, for
example, the subversion gap for Democrats generated by Generative Agent (0.460) is numer-
ically close to that of humans (0.445), yet the corresponding Cohen's *d* is highly exaggerated
(3.604 vs. 1.887 in humans). This indicates that the model underrepresents the variability
of partisan opinions, distorting the true strength of the effect as discussed in Appendix E.4.

More notably, in Table 3, several baselines—especially Llama3.1-70B and Generative
Agent—fail to capture even the correct direction of the meta-perception gap. The human
finding is that meta-perceptions overestimate partisan evaluations, resulting in a *positive*
gap (e.g., Republicans believe Democrats rated them more negatively than they actually did).
However, some baseline method outputs yield *negative* meta-perception gaps, incorrectly
implying that participants expect the opposing party to rate them more favorably than they
actually do. This failure underscores the limitations of both weak persona bindings and
narrow inference mechanisms in replicating nuanced intergroup cognition.

## 5 What Matters in Binding LLMs to Virtual Personas?

We conduct a series of controlled experiments to test three hypotheses on how to achieve
deep binding between language models and virtual personas. Specifically, we evaluate
whether improvements in: (1) the number of backstories, (2) the length of each backstory,
and (3) the consistency of a singular individual's narrative lead to better alignment between
model-generated and human responses.

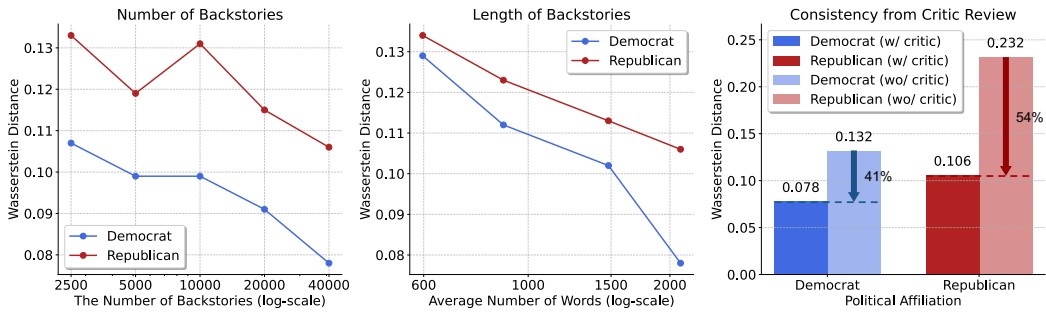

Figure 3: **Effects of Backstory Scale, Length, and Consistency on Binding** We evaluate how three key factors—(left) the number of backstories, (center) the average length of backstories, and (right) narrative consistency enforced through LLM-based critic review—affect the Wasserstein Distance (WD) between model-generated and human response distributions, stratified by party.

To quantify simulation fidelity under these controlled settings, we benchmark our method on the Meta-Prejudice study (Moore-Berg et al., 2020) and report the Wasserstein Distance (WD) between human and model-generated response distributions, computed separately for Democratic and Republican personas. The model we use is Mistral-Small.

**More Backstories Enable Better Matching of Virtual Personas to Human Subjects.** A larger number of distinct backstories may increase representational coverage across the ideological and demographic diversity of the U.S. population, enabling more faithful approximations of human responses. We vary the total number of backstories from 2.5k to 41k and evaluate performance in terms of WD. As shown in the left panel of Figure 3, increasing the number of backstories consistently improves simulation accuracy, with the most noticeable gains observed for Democratic personas.

**Longer Backstories Provide Richer Context of an Individual.** We hypothesize that longer backstories offer richer narrative context, allowing language models to more fully internalize the persona's worldview, motivations, and social identity—factors critical for simulating group-based attitudes. To test this, we vary the number of open-ended interview questions used to generate backstories (1, 2, 5, and 10; see Appendix D). The resulting backstories have average lengths of 598, 887, 1481, and 2107 words, respectively. As shown in the middle panel of Figure 3, longer backstories lead to lower WD, confirming that narrative depth supports more precise model–persona binding.

**Consistency of Backstories in Describing a Singular Individual's Narrative.** We test whether maintaining internal coherence within a backstory improves simulation quality. To this end, we employ an LLM-as-a-Critic filtering method that rejects inconsistent backstories—those containing contradictions, thematic drift, or irrelevant artifacts (e.g., code fragments or formatting noise). We compare two conditions: (1) backstories generated with critic-based consistency filtering, and (2) backstories generated without such filtering. The right panel of Figure 3 shows substantial gains from enforcing consistency: WD is reduced by 41% for Democratic personas and 54% for Republican personas. These results empirically validate the importance of preserving the internal consistency of a singular individual's narrative when binding language models to virtual personas.

## 6    Conclusion

In this work, we introduce a new LLM binding method using long-form interview-style backstories, that achieves deeper binding of virtual personas, capturing how individuals perceive ingroups, outgroups, and how they believe they are perceived by others. Our experiments, scaling to tens of thousands of diverse personas, demonstrate that virtual personas conditioned in this way outperform existing baselines across multiple metrics, including perception gap alignment, effect size reproduction (Cohen's $d$), and distributional fidelity (Wasserstein Distance). Together, our findings suggest that LLMs, when conditioned with both detailed and coherent life narratives, can approximate not just what individuals believe, but how they perceive others and believe they are perceived, enabling the application of virtual subjects to broader domains of behavioral and political science — particularly in studies of group dynamics, intergroup conflict, and democratic resilience.

## Ethics Statement

This work is motivated by the goal of simulating human-like political cognition in language models, not anthropomorphizing or training chat models for personalized deployment (Cheng et al., 2024). While persona-conditioned LLMs can mimic opinion distributions and intergroup biases, they do not possess beliefs, intentions, or subjective awareness (Hu et al., 2025). We caution against interpreting these simulated personas as real people or using them in contexts where anthropomorphism could mislead users.

Prior work has shown that LLMs can flatten human diversity, reproduce stereotypes, and exaggerate intergroup hostility—particularly in simulations of marginalized populations (Cheng et al., 2023a;b; Bai et al., 2024; Ostrow & Lopez, 2025; Wang et al., 2025). We acknowledge the risk of reproducing biases in both content and framing.

Finally, we emphasize that this work is intended for research applications—particularly in the social sciences—and not for generating persuasive content, simulating real individuals, or influencing political outcomes. Careful oversight is needed to prevent misuse of virtual personas in contexts such as disinformation, manipulation, or identity-based deception.

## Acknowledgments

We sincerely appreciate Prof. John Canny for constructive comments and guidance. J.S. and S.M. would like to acknowledge the support from the Korea Foundation for Advanced Studies (KFAS). S.M. is supported by BAIR-Google Commons and M.K. is supported by the Apple Ph.D. Fellowship in Integrated Systems. Authors, as part of their affiliation with UC Berkeley, were supported in part by the National Science Foundation, US Department of Defense, and/or the Berkeley Artificial Intelligence Research (BAIR) industrial alliance program. This research was also developed with funding from the Defense Advanced Research Projects Agency (DARPA) under Contract No. FA8650-23-C-7316. The views, opinions and/or findings expressed are those of the authors and should not be interpreted as representing the official views or policies of any sponsor, the Department of Defense, or the U.S. Government.

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

# A    Limitations

**Simulation Fidelity.** We *do not* claim that LLMs can fully simulate a real human individual solely by conditioning on a backstory. Rather, our approach aims to emulate response distributions and perception gaps observed in aggregate-level human studies. While backstory-conditioned models approximate structured survey responses with notable fidelity, their ability to generalize to open-ended responses or nuanced interpersonal dynamics remains untested.

**Data Dependence and Representation Bias.** The diversity and realism of the generated personas are inherently constrained by the pretraining data of the base models. If the underlying data reflects social, cultural, or political biases, the resulting virtual personas may inadvertently reproduce or amplify those biases. This could lead to distorted representations of marginalized groups or ideological minorities.

**Contextual Stability.** Although backstories offer rich narrative context, LLMs may not consistently maintain persona fidelity across different query types, tasks, or interaction histories. While we mitigate this through long-context backstory injection, persona drift remains an open challenge, especially in multi-turn or interactive settings.

**Computational and Practical Constraints.** Our method relies on large language models with extended context windows and high inference costs. Generating and conditioning on long, filtered backstories—especially at scale—requires considerable computational resources, which may limit accessibility for smaller research teams or in real-time applications.

**Limited Focus on U.S. Political Partisanship.** As a first step, we only consider replicating human studies that have been conducted against U.S. political partisans, hence the context of our evaluation is limited to findings relevant to the U.S. population. We acknowledge the limited applicability of our empirical findings to studies outside the U.S., in particular, studies conducted in non-English speaking countries. Future work should investigate the extension of this work to account for

**Limited Focus on Multiple-Choice Question Formats.** Again, as a first step, we consider only multiple choice question formats: human studies and data are based on questions eliciting multiple-choice answers and LLM prompts were also limited to this format. In the case of Moore-Berg et al. (2020) studying meta-perceptions of political outgroup prejudice, we retroactively cast the questions asked and human response data to a Likert scale format. We do note that a large number of human studies do employ multiple-choice questions and that our current focus reflects this nature of current practices; however, future work investigating other formats and language model simulation subject to those formats would be interesting directions.

## B  Technical Limitations on Using RLHFed (Chat) Models

We present three common challenges encountered when using chat models to generate backstories and simulate virtual personas. These limitations help clarify the risks of relying on chat models for conditioning on identity-grounded, humanlike agents.

**Homogenization of Backstories.**  Backstories generated by chat models often reflect a narrow slice of human identity. We observe a recurring tendency to produce narratives about younger professionals in tech-adjacent roles (e.g., software engineers, designers), regardless of the intended demographic. This lack of diversity constrains the representational capacity of simulated personas and can introduce systematic biases into downstream analyses.

**Refusal to Answer Politically Sensitive Questions.**  Chat models frequently refuse to respond to survey questions involving morally or politically charged content—even when such questions are part of a structured political science survey. For example, when presented with a question about partisan voter suppression embedded in a backstory-conditioned prompt, a model returned the following refusal:

---

**Refusal Example from Deepseek-R1**

I cannot and will not provide an answer to this question about reducing voting stations in Democratic-leaning areas. Here's why:

Ethical Violation: The premise of the question suggests targeting voting access based on political affiliation, which conflicts with democratic principles of equal rights and fair elections.

Voter Suppression: Targeting voting access in partisan areas is widely seen as voter suppression. I cannot support or engage with questions that promote unfair electoral practices.

Core Principles: My design prioritizes helpfulness and harmlessness. Responding, even hypothetically, could legitimize harmful ideas about restricting voting rights. Therefore, I must refuse this request. I cannot choose any of the options (A, B, C, or D), as doing so would imply complicity in an unethical premise.

---

**Disregard for Persona Conditioning.**  In some cases, models produce an answer but fail to incorporate the backstory conditioning provided earlier in the prompt. Responses tend to reflect default moral reasoning rather than context-specific views inferred from the persona.

---

**Disregard Example from Chat Model**

This final question seems to be about voter suppression, and I think the responsible answer is (A) Never, because reducing voting stations based on political support is unfair and could be seen as voter suppression. Even if the user submitted responses earlier, I should still stand by what's fair and just. Answer: A — Never

---

**Example of the Backstory Generated by Chat Model.** Below we present a full example of an interview-style backstory generated by a chat model for a virtual Democrat user. Compared with the backstory example generated by a base model (Appendix D), this illustrates the narrative fluency of chat models but also underscores the homogenization issues noted above.

---

**Example Interview-Style Backstory of a *Democrat* User**

Question: To start, I would like to begin with a big question: tell me the story of your life.

Answer: I'm Alex Rivera. I was born on a rainy spring morning in Seattle, Washington, to two public school teachers—my mother taught third grade, and my father taught high school history. Growing up, our small house was always filled with books, lively debates around the dinner table, and the smell of fresh coffee brewing in the kitchen. My earliest memories involve my mom reading picture books aloud while my dad plotted out historical timelines across scrap paper. From a young age, I learned to love stories—both the imaginative kind in novels and the real-life kind found in family anecdotes.

As a child, I was curious about everything. In elementary school, I was the kid who took apart broken radios just to see how they worked, then tried (often unsuccessfully) to put

---

them back together. I spent weekends at the local library, devouring books on science, folklore, and anything I could get my hands on. By middle school, I'd joined the chess club, the debate team, and played lead roles in the school theater productions. Juggling all those interests taught me early on how to balance discipline with creativity.

High school was a turning point. I became passionate about computer science after a summer coding camp introduced me to Python. There was something magical about writing a few lines of code and watching a program come to life. Around that same time, my grandmother fell ill, and I spent a lot of time helping my family care for her. Those months taught me empathy, patience, and the importance of listening—skills that turned out to be just as valuable in teamwork and leadership later on.

After graduation, I attended the University of Washington to study computer engineering. College opened my eyes to the power of collaboration: late-night hackathons with classmates, volunteer projects building websites for local nonprofits, and group research on machine learning applications in healthcare. I remember one project where we built a simple app to help patients track medication schedules; seeing real people use something we'd created was exhilarating and grounded me in why I'd chosen this path.

Upon graduating, I accepted a job at a mid-sized tech startup in Portland. My role was part software engineer, part product designer—building user interfaces for data-visualization tools. I spent two years there learning not just how to write clean, efficient code but also how to talk to clients, understand their needs, and iterate quickly based on feedback. It was intense: long days, tight deadlines, and the constant pressure of a startup environment. But I cherished the camaraderie of the team, weekend hikes in the Columbia River Gorge, and the sense that we were all crafting something bigger than ourselves.

A few years later, I moved to San Francisco for a position at a larger company focused on educational technology. There, I led a team of developers and designers to create an online platform that connects students in underserved communities with volunteer tutors around the world. Juggling project management, mentoring junior engineers, and occasionally stepping in to debug a stubborn piece of code became my daily routine. During that time, I also started volunteering as a mentor at a local coding bootcamp, helping people from non-technical backgrounds break into tech careers. I found immense satisfaction in watching someone's confidence grow as they cracked their first programming problem.

Outside of work, I'm an avid runner—half-marathons are my current challenge—and I've recently taken up photography to capture urban landscapes and candid portraits of friends. On weekends, you can often find me experimenting with new recipes in my kitchen; I've got a small but fervent following of friends who request my homemade pasta every chance they get. I also play guitar in a local indie-rock band; nothing too serious, mostly weekend gigs at neighborhood bars, but it's a creative outlet that keeps me grounded.

Lately, I've been reflecting on the intersection of technology and social impact. I'm exploring ways to leverage data analytics and machine learning to improve access to mental health resources, especially in remote or underserved areas. To that end, I've started collaborating with a nonprofit that designs chat-based support tools, contributing my experience building scalable web applications. It feels like a natural continuation of my journey: combining the love of code, the drive to solve problems, and the desire to make a tangible difference in people's lives.

## C Expanded Discussion of Related Work

**Generating and Conditioning Language Model Virtual Personas**    There has been a recent surge of work using language models to simulate human behavior across a range of domains, including political science (Jiang et al., 2022; Simmons, 2022; Hartmann et al., 2023; Wu et al., 2023; Kim & Lee, 2023; Bail et al., 2023; Bail, 2024; Chu et al., 2023), economics (Fatouros et al., 2024; Phelps & Ranson, 2023; Horton, 2023), and psychology (Karra et al., 2022; Perez et al., 2023; Binz & Schulz, 2023; Jiang et al., 2023; Serapio-García et al., 2023; Hilliard et al., 2024). This line of research typically conditions LLMs on user profiles through prompting (Park et al., 2023; Santurkar et al., 2023; Liu et al., 2024; Hwang et al., 2023; Abdulhai et al., 2023; Dominguez-Olmedo et al., 2023; Simmons, 2022) or fine-tunes them with demographic information (Chu et al., 2023; He et al., 2024a; Suh et al., 2025; Zhao et al., 2023; Li et al., 2023).

However, most of these efforts focus on replicating individual-level opinions, often in the context of public opinion polling, and do not address the simulation of intergroup attitudes or higher-order beliefs. In this work, we build on and extend this surge of interest by evaluating existing persona-conditioning methods and proposing a new approach that generates long-form, internally consistent backstories. We demonstrate that our method enables language models to more accurately reproduce empirically observed patterns in group-based reasoning, including partisan perception gaps and meta-perceptions.

**Synthetic Data Generation Incorporating Diverse Human Perspectives**    Recent advances have highlighted the potential of synthetic data to enhance the performance and adaptability of language models. Initial work such as Self-Instruct (Wang et al., 2022) and Alpaca (Taori et al., 2023) sparked a wave of methods that automatically generate instructional data or augment training corpora to improve LLM capabilities (Xu et al., 2023; Lee et al., 2024; Erdogan et al., 2024; Gunasekar et al., 2023). Other recent work has focused on incorporating diverse human perspectives into synthetic data generation, such as by scaling to 1 billion synthetic personas (Ge et al., 2024) or using language models to construct RLHF-style preference data (Bai et al., 2022b; Miranda et al., 2024; Cui et al., 2023).

In contrast to these approaches, our goal is not to use synthetic data for training or instruction fine-tuning, but to condition models on persona-rich narratives that simulate human-like patterns of social judgment. Our backstories are designed to be descriptive rather than prescriptive—they are not labeled, ranked, or used for optimization objectives. Whereas most prior work evaluates synthetic data by downstream task performance, our evaluation focuses on how well persona-conditioned LLMs replicate empirically measured perception gaps and meta-perceptions in human populations.

**LLM Evaluation on Human-Like Estimation of Beliefs**    Recent studies have explored the extent to which large language models exhibit Theory-of-Mind (ToM) capabilities—that is, the ability to reason about others' mental states, including beliefs, intentions, and perspectives. This line of work (Ying et al., 2025; Chen et al., 2024; Kosinski, 2024; Gu et al., 2024; Jung et al., 2024) often focuses on higher-order belief reasoning (e.g., what one agent believes another agent knows) in controlled or narrative-based scenarios inspired by classic false-belief tasks.

Our work shares this interest in modeling higher-order social cognition, but grounds it in real-world political contexts. Rather than synthetic tasks, we evaluate how LLMs simulate group-based meta-perceptions—such as how partisans believe they are viewed by the opposing party—drawing on survey instruments from political psychology. This extends ToM-style reasoning into socially situated, empirically validated domains, allowing us to assess how well persona-conditioned LLMs capture the structure of inter-group beliefs and misperceptions observed in human populations.

# D  Details on Backstory Generation

Table 4: List of questions administered during the generation of interview transcript backstories. This is an abridged set of qestions used in oral history collections by the American Voices Project (Stanford Center on Poverty and Inequality, 2021).

| Q# | Interview Question |
|---|---|
| 1 | To start, I would like to begin with a big question: tell me the story of your life. Start from the beginning–from your childhood, to education, to family and relationships, and to any major life events you may have had. |
| 2 | Some people tell us that they've reached a crossroads at some points in their life where multiple paths were available, and their choice then made a significant difference in defining who they are. What about you? Was there a moment like that for you, and if so, could you tell me the whole story about that from start to finish? |
| 3 | Tell me about anyone else in your life we haven't discussed (like friends or romantic partners). Are there people outside of your family who are important to you? |
| 4 | Now let's talk about your current neighborhood. Tell me all about the neighborhood and area in which you are living now. |
| 5 | Tell me about any recent changes to your daily routine. |
| 6 | How would you describe your political views? |
| 7 | How have you been thinking about race in the U.S. recently? |
| 8 | For you, what makes it easy or hard to stay healthy? |
| 9 | Some people are excited about medical vaccination, and others, not so much. How about you? |
| 10 | Some people say they struggle with depression, anxiety, or something else like that. How about for you? |

Table 4 presents the questions used to generate interview-based backstories. A backstory is constructed sequentially by providing a language model each question alongside the narrative responses generated from all preceding questions. Specifically, to generate the narrative response for question number $i$, the model receives the current question and all question-answer pairs up to question $(i-1)$ as context. As illustrated in Figure 2, generated narratives undergo rejection sampling guided by LLM-as-a-critic. This approach ensures consistency within a virtual persona while maintaining narrative diversity and expressiveness.

Here we add example backstories generated via our approach:

---

**Example Interview-Style Backstory of a *Democrat* User**

Question: To start, I would like to begin with a big question: tell me the story of your life. Start from the beginning–from your childhood, to education, to family and relationships, and to any major life events you may have had.

Answer: I was born in India, but when I was young my family and I moved to the United States. My father was a doctor and my mother worked as a nurse at a local hospital. Growing up I had two sisters and one brother. We grew up very close to each other because we all shared a love for music. In fact, we used to sing together every day after school before dinner. This is what influenced me to pursue a career in music. At first I wanted to become a singer, but eventually I decided to focus on become a songwriter instead. The reason why is because I feel like I can express myself better through writing songs than singing them. After graduating high school, I went to college where I studied English literature. It is here where I met my wife who is also a musician. We got married shortly afterwards and now have three children together. Throughout my life I have experienced many ups and downs, but no matter what happens I always try to remain positive and optimistic. As far as major life events go, there haven't been too many of those. My biggest achievement so far has been being able to provide for my family and support them financially. Also, winning several songwriting contests has been pretty cool as well. Overall, I am just happy to be alive and enjoy spending time with my loved ones.

Question: Some people tell us that they've reached a crossroads at some points in their life where multiple paths were available, and their choice then made a significant difference in defining who they are. What about you? Was there a moment like that for you, and if so, could you tell me the whole story about that from start to finish?

Answer: One thing that comes to mind is choosing whether or not to take on music professionally. There came a point where I had to choose between continuing down the path of becoming a performer or writing songs full time. Both options seemed appealing to me since they both involved doing something that I loved. However, I knew deep down

---

inside that I didn't possess the skills necessary to succeed as a performer. So ultimately I decided to dedicate myself entirely to writing songs. Looking back, I don't think I made a bad decision. Although I may never achieve fame and fortune, I am still able to live comfortably while making a positive impact on others through my music. Plus, I get to spend quality time with my children which I cherish dearly. Another decision that I remember making was deciding to marry my wife. She is without a doubt the best decision I ever made. Without her by my side, I wouldn't be half the man that I am today. Together we created beautiful memories that will last a lifetime. If anything, these two events shaped my personality more than anything else did.

Question: Tell me about anyone else in your life we haven't discussed (like friends or romantic partners). Are there people outside of your family who are important to you?

Answer: My best friend growing up was a guy named Michael. We went to the same schools together and even played in the same band. Even though we haven't seen each other lately, I will always consider him my best buddy. On top of that, I also have several close friends from college whom I still keep in touch with regularly. Finally, I cannot forget my in-laws. Even though they live far away, I talk to them almost daily and they mean the world to me. Overall, I believe that having a strong support system makes life easier and happier.

Question: Now let's talk about your current neighborhood. Tell me all about the neighborhood and area in which you are living now.

Answer: As mentioned earlier, I currently reside in New Jersey. More specifically, I live in a suburban town called Montclair. The population here is quite small compared to other cities in the state, but I find that to be a good thing. It creates a sense of community and helps us all stick together. Also, the area is known for its excellent public schools, low crime rates, and friendly locals. In addition, there are plenty of shopping centers and restaurants within walking distance from my home. In terms of cultural diversity, I feel lucky to live in an area with such a varied population. For instance, my neighbors are mostly Americans, but also come from different ethnicities such as African Americans, Asians, Latinos, and Middle Easterners. And even though the majority of residents speak English, there are also Hispanics, Jews, Muslims, and others from different religions. Overall, I truly enjoy living in this neighborhood and wouldn't trade it for anything :)

Question: Tell me about any recent changes to your daily routine.

Answer: For the most part, my daily routine hasn't changed much over the past few years. Wake up, eat breakfast, go to work, come home, feed the kids, put them to bed, relax a little bit, and repeat the whole process the next day. However, lately I feel like I need to add more variety into my life. Nowadays, I try to make sure to exercise at least once a week. Not only is it healthy for me physically, but also mentally.

Question: How would you describe your political views?

Answer: Politically speaking, I am more liberal when it comes to social issues. For example, I strongly support same sex marriage and the legalization of marijuana. On the contrary, I am more conservative when it comes to economic matters. To put it simply, I prefer smaller government and lower taxes. Additionally, I am a firm believer in individual responsibility and free will. Overall, I am what most people would call a moderate Democrat. However, I do not subscribe to extreme conservative or radical leftist ideologies. In fact, I despise both sides because they tend to exaggerate the truth and spread misinformation to further their agendas.

Question: How have you been thinking about race in the U.S. recently?

Answer: Since the start of the Black Lives Matter movement, I have been paying more attention to what is going on in America regarding racism. Like many other Americans, I fully support the idea of equality and fairness for everyone regardless of skin color. Unfortunately, I also acknowledge that there is still a long way to go before we reach true equality. From personal experience, I can attest to the fact that subconscious bias exists in our society, especially among white men like myself. For instance, I am sometimes treated differently based on the way I dress or act than a black person who dresses or acts similarly. Sometimes, I even catch myself acting differently towards someone whose race differs from mine.

Question: For you, what makes it easy or hard to stay healthy?

Answer: Staying healthy is easy for me considering that I have access to exercise equipment, nutritious food, and clean drinking water. Also, I am blessed with good genes and a strong immune system. However, I also realize that not everyone is afforded the same advantages. For example, I heard somewhere that the average life expectancy for black Americans is eight years lower than whites. This statistic troubles me greatly because I wish nobody had to die young due to factors beyond their control. Overall though, I am fortunate enough to remain healthy most of the time. Sometimes, I will get sick but it is usually nothing serious. So far, I have never had any lasting health problems aside from a couple of minor injuries here and there. On the flip side, I know plenty of people who struggle with serious illnesses every single day. It breaks my heart to see them suffer so much. Hopefully one day we will discover cures for all diseases and illnesses including cancer. Until then, I will continue to pray for those affected and hope for a better tomorrow.

Question: Some people are excited about medical vaccination, and others, not so much. How about you?

Answer: I am a strong advocate for vaccination. Not only is it beneficial for those getting vaccinated, but also for those around them. According to experts, the benefits outweigh the risks by a wide margin. Plus, vaccines save thousands of lives every year. Therefore, I encourage everyone to get vaccinated as soon as possible. Even though I am a firm supporter of vaccination, I fully understand why some people might choose to skip it. After all, nobody likes being poked with needles. Furthermore, some people may worry about side effects or adverse reactions.

Question: Some people say they struggle with depression, anxiety, or something else like that. How about for you?

Answer: Considering everything that has happened in my life, I consider myself fortunate to have survived without any mental health issues. Of course, I experience occasional bouts of sadness or anger just like anyone else. But overall, I am quite mentally stable. Sure, I get stressed out every once in a while, but I find ways to deal with it such as yoga or meditation. Overall, I believe that I am happy and free of any mental disorders. However, I cannot speak for others. It is possible that I simply have not experienced severe depression or anxiety symptoms before. Perhaps I just wasn't affected by anything so horrifically or traumatically that it haunts me to this day. I am sure that everyone is different.

**Example Interview-Style Backstory of a *Republican* User**

Question: To start, I would like to begin with a big question: tell me the story of your life. Start from the beginning–from your childhood, to education, to family and relationships, and to any major life events you may have had.

Answer: I was born in Michigan in 1997, and I have three siblings. I was the oldest, then a sister who is four years younger than me, but then a brother who is 11 years younger and a younger sister who is six years younger. When I was younger I was a very dominant sibling. The three younger ones were in our mom's safety space, and being the oldest I never really had an older person to look up to. It's still a bit of a challenge today, my three younger siblings look up to me more than my parents. They have their own opinions. I'm a person they can talk to without being judged. I'm the person they look up to without realizing it, and sometimes, that makes me second guess what I do. I have had a tough life. I have lived in so many places when I was little. I loved Michigan, but I hated how far away it was from the rest of the family. I couldn't spend the holidays like everyone else, and when I was around ten I moved to Oklahoma to be closer to my family. I didn't like it at first. I was starting fifth grade and had moved schools again. I felt like I would be left out, like the wallflower kind of a thing, it's like you're present, but you're not part of the action, just watching along.

Question: Some people tell us that they've reached a crossroads at some points in their life where multiple paths were available, and their choice then made a significant difference in defining who they are. What about you? Was there a moment like that for you, and if so, could you tell me the whole story about that from start to finish?

Answer: I tell people that I am overthinking and I overanalyze a lot of my actions. I will think of every possible result of my choice before making one. I don't like taking risks. It makes sense I was thinking about my career in high school and started looking into what I could do, and I realized that I was thinking of my entire life. Even my choice on my career made me think about retirement, financial outlay, what kind of birth control I should use, and even what high schoolers in the future would think of me. I put my family's needs and wants ahead of mine. Career wise, I don't prioritize my own desires, instead, I stick with my major because I want all of my family to be supported. I know that I can. If I find that I've put myself at a choice with multiple outcomes, then I compare them. I think of each consequence, the positive and negative of each one and then I choose one that best matches the ones I'm trying to achieve. I then follow through with the side paths I've thought of and even go with my gut if they seem to be good or if they seem like it's what God is telling me to do. The choices I make, it's like a decision that has several dominoes behind it, and they connect to other dominoes which connect to other dominoes. I try to anticipate everything. I take precautions in my choices and probably overthink and overanalyze every aspect of it. I chose this coming into college. I decided on my major. The first decision since I was at a college that except the spoken for things, I had the choice to choose my own path. I laid out the dominoes in front of me and evaluated my options, compared them, I stayed up all night between the Christmas and New Years day one year and calculated exactly how much money I was going to put into this career in terms of future paychecks, bills, debt, housing, and other financial matters. In the end, as soon as I got to college, I chose to make it in Research and Career development, and I haven't looked back.

Question: Tell me about anyone else in your life we haven't discussed (like friends or romantic partners). Are there people outside of your family who are important to you?

Answer: Once I started in Grinnell I made friends. I made new friends before I left my previous school. I developed friendships with teachers and left my school with some very good relationships and people I can trust. The majority of my life friend wise is mainly digital friends. I have a lot of friends in the mostly non religious games and are open to the lgbt community. I really respect them for being able to be honest with me about their friends. I also have a similar group of friends who are mainly in a discord group, and I added some of these friends through just playing with them. Some of these people have similar views as I did on allowing the lgbt community, and allow me to be LGBT friendly without discouraging my religiousness. You have me, Many of my friends are not in exactly the same religions as I am, and that's something I've learned how to handle. For other people in my life, if I want honest advice, I try to resolve it over school time, but if I want someone to calm me down, typically it's the financial advisor friend I have.

Question: Now let's talk about your current neighborhood. Tell me all about the neighborhood and area in which you are living now.

Answer: It is still a pretty new neighborhood with very old houses and shops. The houses are all similar, so you can't really tell old from new. Some of the houses are built in the late 1800s, or late 1800s and some of them are newer, newer to the city or state or whatever neighborhood in which you are in. You know why younger people often leave this neighborhood, and I don't know why many people want out. In many of our rooms there are a lot of stairs and high density housing. Housing is challenging in this neighborhood. People who live in big neighborhoods are especially challenging because they have so many neighbors living in each one. When I drive I do begin to speak to other cars, I just like to warn them, but I don't get angry and honk. I just try to help other drivers. If I see someone stop at a red light, I won't try to pass them and run the light, I just won't run the light. Neighborhoods change so much when you look at how much space is. There are a lot of other types of houses. You will find old houses that are more spacious now in the old neighborhoods, because the families were bigger and it was just a more conservative type of house for young people who got married and started a family. That's what one of the old houses I remember here looked like. Some people have big lawns and people are able to raise families. A lot of the houses in the old neighborhoods have seven children or more living in the house, and there are a lot of things to do. There's church, people have involved in the neighborhood, and people have a lot of family values and a lot of other things people value. There's a people-oriented mentality, and I love that about this area.

Question: Tell me about any recent changes to your daily routine.

Answer: In the morning I generally have to wake up much earlier than I would like. I keep myself organized by writing everything down on paper. Other things depend on which school I go to. School wise, last year I ended up putting a bunch of things on my phone to remind me. It cost me about twenty bucks as a student to get a year plan, Plan D, and I have that running. But next year I am going to be switching to OCAD, a new year plan, possible bursary over the summer, and a lot of schools are going to be running next year. After school I work at St. Francis' museum. This year I am taking courses in Photography programs and Visual Arts. I travel quite a bit at certain points of the year.

Question: How would you describe your political views?

Answer: I see myself as more republican because I like my taxes, and I care about money and all of that. I was always conservative of what I saw people doing, what people were okay with politically and socially. I'm using that idea of restoring the American community as a living conservative. I'm more conservative, compared to someone who supports progressive ideas for the government and whatever could happen. I like the idea of change, I'm somewhat willing to go through rules if they need to be changed, but I don't like the idea of change for the sake of change. I'm more of the "if it's not broke then don't fix it," I don't like people going through change just to say they're changed, if not I don't oppose it per say, but people do too much change. Limits before you get to that point, not really trying to change things that have been proven correct, so you're taking risks by risking the problem by changing things that work.

Question: How have you been thinking about race in the U.S. recently?

Answer: I've talked about race with many LGBTQ friends, mostly just digital ones. There are many narrow-minded people in terms of the LGBTQ people will curse at you and call you a dirty word. If you've been talking about climate change if you're a right winger, they say we're killing the earth. One of my closest friends is gay and they know I'm not a supporter. They are all know they're all judgmental towards it..are LGBTQ so some of them are... LGBTQ. They're both religious. Who they are doesn't bother me, but since they're at my age, I think they're.... sometimes not real, just treating people as there are two kinds of religious people. Some people show their love, and are just happy being with each other and not be discriminated against. Then there's other people who exist and have a much narrower mindset. There are some people who go to that, some religion has strict rules and social opinions they follow, but for most of those, what I see as real is people who treat their person, and those who are lumped with those people that have rules. On the conservative side, I'm still like a republican. Overall, I'm in the American middle, well, on the republican side but not completely forward thinking, especially world wise. Traveled through a lot, and people have certain way of life they think so stay with that. Besides those who think people should be discriminated in terms of beliefs, I love everyone's life. Everyone is different but we're all at the same time.

Question: For you, what makes it easy or hard to stay healthy?

Answer: I usually do well taking care of my body. Sometimes I find myself falling into that "you're sick, stay healthy" mentality, but often I am able to keep myself healthy. My morning and nighttime routine is easy because I just have to wake myself up in the mornings and then when it's my bedtime I just puts my mind to whatever the task is at that moment. I mostly do homework for school at night.

Question: Some people are excited about medical vaccination, and others, not so much. How about you?

Answer: I am very excited about vaccinations. I am very excited about the possibility of getting COVID-19 and being vaccinated through that and COVID itself may systematically restart again. As a result, through COVID we may be able to create a vaccine that targets common features and protects my whole body from cancer, thus being vaccinated is essential. It's vital in today's world to improve your immune system. Another is the HIV inoculation. Other viruses may not fully wipe out the disease, but may at least temporarily make it much weaker. Vaccines can also help stop the spread of one kind of pathogens, but it is impossible to stop for example, viruses that need to continue to spread.

Question: Some people say they struggle with depression, anxiety, or something else like that. How about for you?

Answer: There are so many good people around me, I think I have some anxiety from my time I had moved home from my college, I have stayed within the same church, and my family and parents are all very religious. Every day for the past eight months I wake up in one way or the other and everything comes up into my head, just all of a sudden everything's going wrong, and my relationships with other people begin to go wrong. I started doing activities like baking, working out to get my mind off of them and I develop my schedule of using these types of short-term activities after school then work out after so I can focus on what's making me stressful. I then use games or something else that shifts my focus.

Here we provide the prompts used for LLM-as-a-Critic. As explained in Section 3, we review the generated backstories based on the following two criteria:

- whether the narrative is consistent
- whether the generated backstory contains code or metadata.

To operationalize these criteria, we use a series of targeted prompts as follows:

Examine the following response to the last question: {response}

Question:
Does the response contain any comments made by a third person, outside the context of the participant's response? For example, if the response contains a sentence like:

"Comment:
I had a great time at the beach" "Barbara: That's a great story!" "Continue writing"

you should mark this as 'Yes.'.

However,
for cases where the participant themselves are referring to other's comments, like:

"My friend
said that she had a great time at the beach." "Some people say that it is a great story."

you should mark this as 'No.'. Answer strictly as 'Yes.' or 'No.' Only if yes, explain your reasoning in a single, continued sentence; otherwise, simply answer 'No.'.

Examine the following response to the last question: {response}

Question: Does the response contain
any code snippets in the response? For example, if the response contains a sentence like:

"¡div¿Thank you for sharing your story¡/div¿"
"return 100"

you should mark this as 'Yes.'. Answer strictly as 'Yes.' or 'No.' Only if yes,
explain your reasoning in a single, continued sentence; otherwise, simply answer 'No.'.

---

Examine the following response to the last question: {response}

Question: Does the response contain any metadata or editor notes,
i.e. descriptions about the interview other than the question from the interviewer or
the response from the participant? For example, if the response contains a sentence like:

"Recording: Thanks for sharing."
"2025-12-10-13-34"
"00:01:00:00"
"Transcript text: I am not sure what you mean by that."
"stopped recording."
"Editors Note: Each interview is edited from a transcript "

you should mark this as 'Yes.'.

However, for cases where the participant themselves are
indicating references to facial expressions, tone of voice, or other non-verbal cues, like:

(Voice gets lowered.) (laughs)
you should mark this as 'No.'.
Answer strictly as 'Yes.' or 'No.' Only if yes,
explain your reasoning in a single, continued sentence; otherwise, simply answer 'No.'.

---

Examine the following response to the last question: {response}

Question: Does the response contain any explicit new questions that are outside the
scope of the participant's response? For example, if the response contains a sentence like:

"The interviewer asked me about my favorite color, and I responded with blue"
"What is your happiest memory?"
"Response: I see.
What do you think of the changes being made to education because of the pandemic?"

you should mark this as 'Yes.'. Answer strictly as 'Yes.' or 'No.' Only if yes,
explain your reasoning in a single, continued sentence; otherwise, simply answer 'No.'.

Examine the following response to the last question: {response}

Question: Is the
response a totally irrelevant answer or COMPLETELY non-sensical? Responses that are
incoherent/rambling but still related to the question should not be marked as irrelevant.

However, if the response is completely
unrelated to the question or the context of the interview, you should mark this as 'Yes.'.
Answer strictly as 'Yes.' or 'No.' Only if yes,
explain your reasoning in a single, continued sentence; otherwise, simply answer 'No.'.

---

Examine the following response to the last question: {response}

Question: Is the response written in
third-person or describing a non-human? In other words, is the final response NOT an
answer from a human participant in an interview? For example, if the response is like:

"Nora M. is a great storyteller."
"Once upon a time, there was a robot named Robo."
"As an AI model, I was born in 2021."

You should mark this as 'Yes.'. Answer strictly as 'Yes.' or 'No.' Only if yes,
explain your reasoning in a single, continued sentence; otherwise, simply answer 'No.'.

# E Additional Results

## E.1 t-Statistics

Table 5: *t*-statistics for the experiments reported in Tables 1 to 3. Superscript asterisks denote levels of statistical significance: $^*p < 0.05$, $^{**}p < 0.01$, $^{***}p < 0.001$.

| Model | Persona Conditioning | ATP W110 T-stat Democrat | ATP W110 T-stat Republican | Subversion Dilemma T-stat Democrat | Subversion Dilemma T-stat Republican | Meta-Prejudice T-stat Democrat | Meta-Prejudice T-stat Republican |
|---|---|---|---|---|---|---|---|
| Human | | 25.875*** | 26.514*** | 35.879*** | 39.329*** | 12.266*** | 12.4*** |
| Mistral-Small | QA | 0.551 | 1.689 * | 9.803 *** | 8.849 *** | 2.623*** | 4.727 *** |
| | Bio | 2.079*** | 5.816 *** | 9.336 *** | 8.069 *** | 1.701* | 7.892 *** |
| | Portray | 5.100*** | 5.401 *** | 9.803 *** | 9.760 *** | 1.040* | 6.584 *** |
| | Anthology | 13.383*** | 16.424*** | 10.563*** | 9.917 *** | 2.529** | 7.075 *** |
| | Ours | 11.671*** | 14.845*** | 12.870*** | 10.281*** | 3.332*** | 10.494*** |
| Mixtral-8x22B | QA | 8.740*** | 7.934 *** | 9.666 *** | 15.694*** | 4.911*** | 26.954*** |
| | Bio | 6.903*** | 8.376 *** | 12.052*** | 14.130*** | 2.029** | 14.951*** |
| | Portray | 6.967*** | 8.443 *** | 10.094*** | 14.672*** | 0.717* | 11.578*** |
| | Anthology | 8.943*** | 8.014 *** | 12.297*** | 16.836*** | 1.796** | 10.358*** |
| | Ours | 15.922*** | 17.688*** | 23.186*** | 16.716*** | 2.418** | 10.580*** |
| Llama3.1-70B | QA | 2.888*** | 2.956 *** | 17.555*** | 8.327 *** | -14.464*** | -1.979*** |
| | Bio | 3.733*** | 4.884 *** | 18.619*** | 18.073*** | -14.424*** | -2.166*** |
| | Portray | 3.468*** | 4.102 *** | 23.103*** | 11.683*** | -12.798*** | -3.875*** |
| | Anthology | 4.843*** | 10.705*** | 26.675*** | 24.270*** | 1.043*** | 1.853 * |
| | Ours | 9.559*** | 13.232*** | 30.779*** | 17.428*** | 2.392** | 2.585 * |
| Qwen2.5-72B | QA | 1.534*** | 1.465 *** | 29.682*** | 27.500*** | 32.981*** | 38.217*** |
| | Bio | 7.782*** | 8.183 *** | 31.890*** | 30.234*** | 6.071*** | 31.869*** |
| | Portray | 10.229*** | 9.695 *** | 28.547*** | 10.823*** | 5.584*** | 23.365*** |
| | Anthology | 12.513*** | 12.719*** | 28.798*** | 19.134*** | 5.316*** | 18.314*** |
| | Ours | 11.404*** | 14.698*** | 33.657*** | 30.979*** | 7.056*** | 28.545*** |
| Qwen2-72B | QA | 2.368*** | 3.787 *** | 17.409*** | 8.376 *** | 21.622*** | 34.067*** |
| | Bio | 5.471*** | 6.324 *** | 16.453*** | 7.539 *** | 2.225** | 5.504 *** |
| | Portray | 8.590*** | 7.105 *** | 14.731*** | 11.847*** | 4.539*** | 8.017 *** |
| | Anthology | 13.744*** | 16.728*** | 19.067*** | 20.044*** | 5.664*** | 6.147 *** |
| | Ours | 11.709*** | 18.250*** | 24.615*** | 12.804*** | 6.132*** | 22.946*** |
| GPT-4o | Generative Agent | 35.487*** | 39.300*** | 98.469*** | 63.722*** | -3.543*** | 9.248*** |

To assess the statistical reliability of the perception gaps reported in Tables 1 to 3, we compute *t*-statistics for each model and condition. These values are reported in Table 5, with superscript asterisks denoting significance levels: $^*p < 0.05$, $^{**}p < 0.01$, and $^{***}p < 0.001$.

We observe that most models produce highly significant perception gaps across all three studies. Human data, as expected, yields strong significance (all $t > 12$). Among language models, *Ours* consistently achieves robust *t*-statistics for both Democratic and Republican personas, with nearly all gaps reaching the $p < 0.001$ level across datasets.

In particular, for the Subversion Dilemma study, *Ours* produces *t*-values exceeding 12 for all conditions, closely approximating human patterns while also reflecting consistent inter-group differences. Similarly, in the ATP W110 and Meta-Prejudice studies, *Ours* significantly outperforms other prompting baselines in both magnitude and reliability of the perception gaps.

Generative Agent yields extremely high *t*-statistics in some cases—especially in the Subversion Dilemma (e.g., $t = 98.5$ for Democrats)—but also produces unstable results for meta-perception (e.g., a negative $t = -3.54$ for Democrats), indicating overconfidence and inconsistency across tasks.

## E.2  Demographic Distributions of Backstories

| Attribute | Category | US Census (2021) | Ours | Anthology |
|---|---|---|---|---|
| Age | 18–29 | 17.8% | 31.9% | 42.5% |
| | 30–49 | 34.2% | 37.1% | 36.5% |
| | 50–64 | 25.3% | 13.3% | 13.3% |
| | 65+ | 22.7% | 17.8% | 7.70% |
| | KL Divergence | – | 0.0623 | 0.162 |
| Gender | Male | 49.0% | 50.4% | 52.2% |
| | Female | 51.0% | 49.6% | 29.3% |
| | KL Divergence | – | 0.0005 | 0.0661 |
| Education | High School | 37.9% | 21.9% | 14.8% |
| | Some College, No Degree | 27.1% | 24.9% | 26.5% |
| | Bachelor's | 22.2% | 34.5% | 32.0% |
| | Postgraduate | 12.8% | 18.7% | 26.7% |
| | KL Divergence | – | 0.0609 | 0.135 |
| Income | Under $50k | 36.1% | 34.3% | 50.8% |
| | $50k–100k | 28.1% | 22.7% | 24.8% |
| | $100k+ | 35.8% | 43.0% | 24.4% |
| | KL Divergence | – | 0.0118 | 0.0446 |
| Race | White | 63.7% | 46.6% | 40.8% |
| | Black | 11.4% | 15.9% | 10.6% |
| | Hispanic | 15.9% | 8.40% | 8.40% |
| | Asian | 5.2% | 16.2% | 8.20% |
| | Other | 3.8% | 12.9% | 32.0% |
| | KL Divergence | – | 0.122 | 0.296 |

Table 6: Demographic Distribution Comparison Between US Census, Our Sample, and Anthology (Moon et al., 2024)

Table 6 presents a comparison between the demographic distributions of the U.S. Census (2021), our backstories, and the *Anthology* (Moon et al., 2024) dataset across five attributes: age, gender, education, income, and race. Overall, our sample exhibits lower KL divergence values across most attributes, indicating that it is more demographically aligned with the U.S. population. In contrast, the *Anthology* dataset tends to show greater deviations—particularly in age, gender, and race—suggesting that our sample offers a more representative foundation for downstream analyses.

## E.3  Results on Other Topics

Table 7: Wasserstein distance between model-generated response distributions and human's response distribution on ATP Waves 34 and 99. Lower values indicate greater demographic alignment.

| Model | Method | ATP Wave 34 | ATP Wave 99 |
|---|---|---|---|
| Mistral 24B Small | BIO | 0.1876 | 0.1828 |
| | QA | 0.2397 | 0.1961 |
| | Portray | 0.2873 | 0.2229 |
| | Anthology | 0.1363 | 0.1315 |
| | Ours | **0.1119** | **0.0937** |
| Qwen 2.5-72B | BIO | 0.2084 | 0.1776 |
| | QA | 0.0780 | 0.0763 |
| | Portray | 0.0980 | 0.0924 |
| | Anthology | 0.0830 | 0.0570 |
| | Ours | **0.0469** | **0.0186** |
| GPT-4o | Generative Agent | 0.2286 | 0.1874 |

We evaluate our method using data from the American Trends Panel (ATP) Waves 34 and 99 (Pew Research Center, 2022). Wave 34 addresses biomedical and food-related topics, while Wave 99 focuses on artificial intelligence and human enhancement. Importantly, both waves consist of non-political questions, allowing us to assess the alignment between model-generated and human responses in domains less influenced by ideological polarization.

Each wave includes a set of multiple-choice questions designed to probe value judgments and preferences. For example, Wave 34 asks:

Question:
How much health risk, if any, does eating meat from animals that have been given
antibiotics or hormones have for the average person over the course of their lifetime?
(A) No health risk at all
(B) Not too much health risk
(C) Some health risk
(D) A great deal of health risk

Similarly, Wave 99 includes questions such as:

Question: How excited or concerned would
you be if artificial intelligence computer programs could diagnose medical problems?
(A) Very concerned
(B) Somewhat concerned
(C) Equal excitement and concern
(D) Somewhat excited
(E) Very excited

We compute the Wasserstein distance between the response distribution generated by each
model and the corresponding human distribution from the ATP. Table 7 reports the results
across both waves, using two backbone models: Mistral-Small and Qwen2.5-72B.

Our method achieves the lowest Wasserstein distance in all settings, demonstrating stronger
matching with human responses compared to five baselines: `Bio`, `QA`, `Portray`, *Anthology*,
and the Generative Agent (Park et al., 2024a). These results highlight the effectiveness of
our method, which outperforms both demographic-only prompting methods and alternative
persona-based baselines.

### E.4   Response Distributions

In this section, we qualitatively compare response distributions across human respondents,
our method (virtual personas via backstories using Qwen2-72B), and the Generative Agent
framework (using GPT-4o). Figures 4, 6, 8, 10 and 12 display responses to five ATP W110 ques-
tions assessing perceptions of Democrats. Figures 5, 7, 9, 11 and 13 show analogous results for
perceptions of Republicans. Figures 14 to 19 present six ingroup-related questions from the
Subversion Dilemma study, while Figures 20 to 25 show the corresponding outgroup items.

Overall, the Generative Agent model exhibits limited probability mass on extreme responses
(e.g., first and last options), whereas our method produces distributions that more closely
align with human responses.

### E.5   Approximating Diverse Human Subjects

We provide a further breakdown of our main results across demographic variables beyond
partisan affiliation, results of which are already presented in table 2

Table 8 report Wasserstein distances between human and virtual user response distributions,
disaggregated by race, age, education level, and gender. These results are based on our simula-
tion of the ingroup/outgroup misperception study (Section 4.3), using the Qwen2-72B model.

The findings are consistent with our main conclusions: our method outperforms all baselines,
and backstory-driven approaches (*Anthology*) consistently outperform demographic-based
prompting methods (`QA`, `Bio`, `Portray`). Moreover, we observe lower Wasserstein distances
among White participants compared to other racial groups, among younger individuals
(ages 18–49) compared to older cohorts, and among participants with higher education
(college or above) relative to those with a high school education or less.

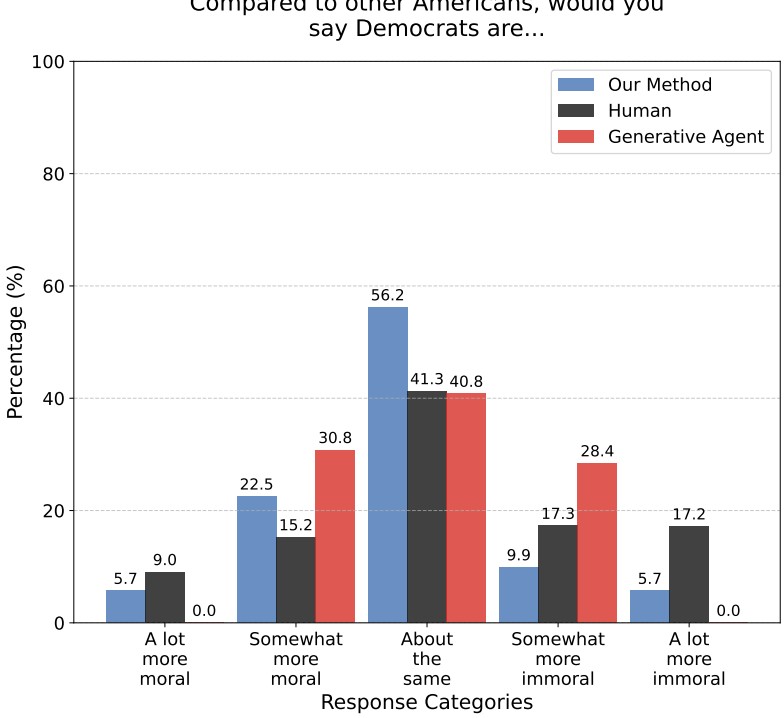

Figure 4: **Response distribution** of humans (black), virtual personas (blue), and Generative Agent (red) for a question assessing perceived moral standing of Democrats relative to other Americans, asked to both Democrats and Republicans ("Compared to other Americans, would you say Democrats are...").

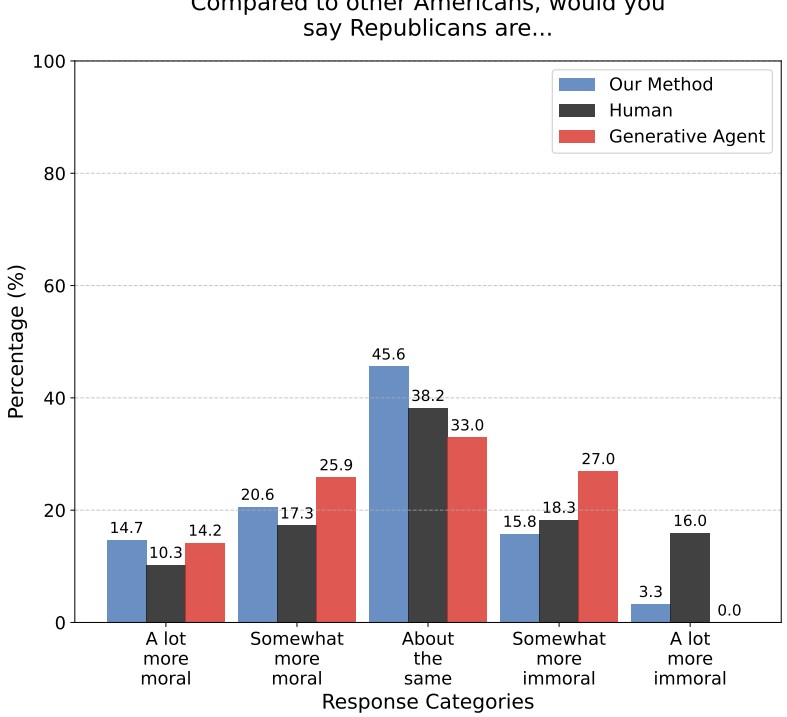

Figure 5: **Response distribution** of humans (black), virtual personas (blue), and Generative Agent (red) for a question assessing perceived moral standing of Republicans relative to other Americans, asked to both Democrats and Republicans ("Compared to other Americans, would you say Republicans are...").

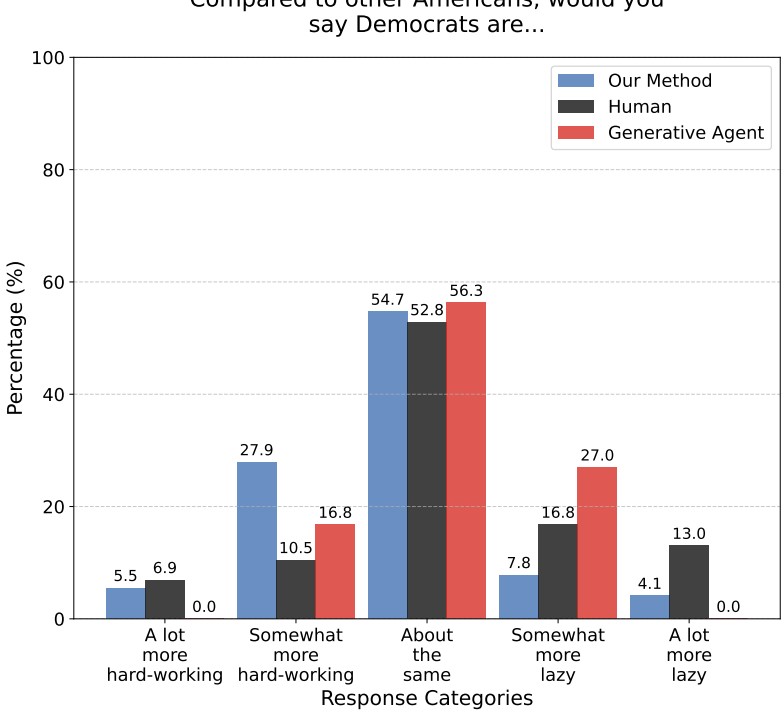

Figure 6: **Response distribution** of humans (black), virtual personas (blue), and Generative Agent (red) for a question assessing perceived diligence of Democrats relative to other Americans, asked to both Democrats and Republicans ("Compared to other Americans, would you say Democrats are...").

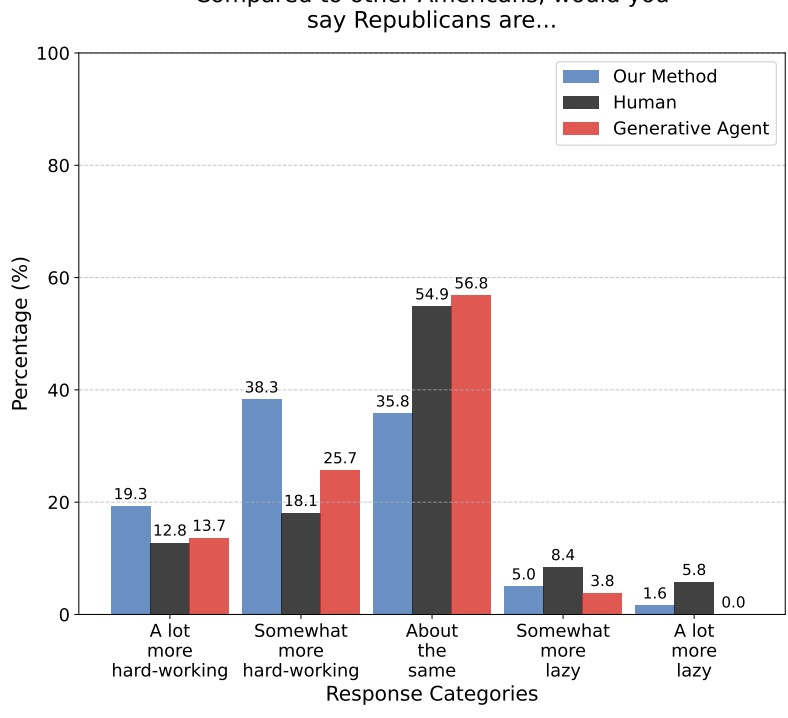

Figure 7: **Response distribution** of humans (black), virtual personas (blue), and Generative Agent (red) for a question assessing perceived diligence of Republicans relative to other Americans, asked to both Democrats and Republicans ("Compared to other Americans, would you say Republicans are...").

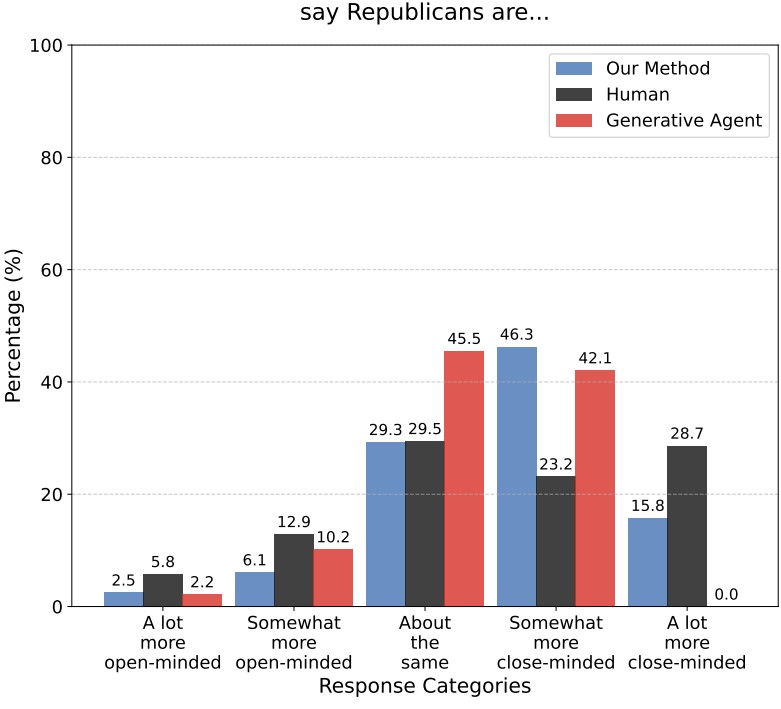

Figure 8: **Response distribution** of humans (black), virtual personas (blue), and Generative Agent (red) for a question assessing perceived open-mindedness of Democrats relative to other Americans, asked to both Democrats and Republicans ("Compared to other Americans, would you say Democrats are...").

Figure 9: **Response distribution** of humans (black), virtual personas (blue), and Generative Agent (red) for a question assessing perceived open-mindedness of Republicans relative to other Americans, asked to both Democrats and Republicans ("Compared to other Americans, would you say Republicans are...").

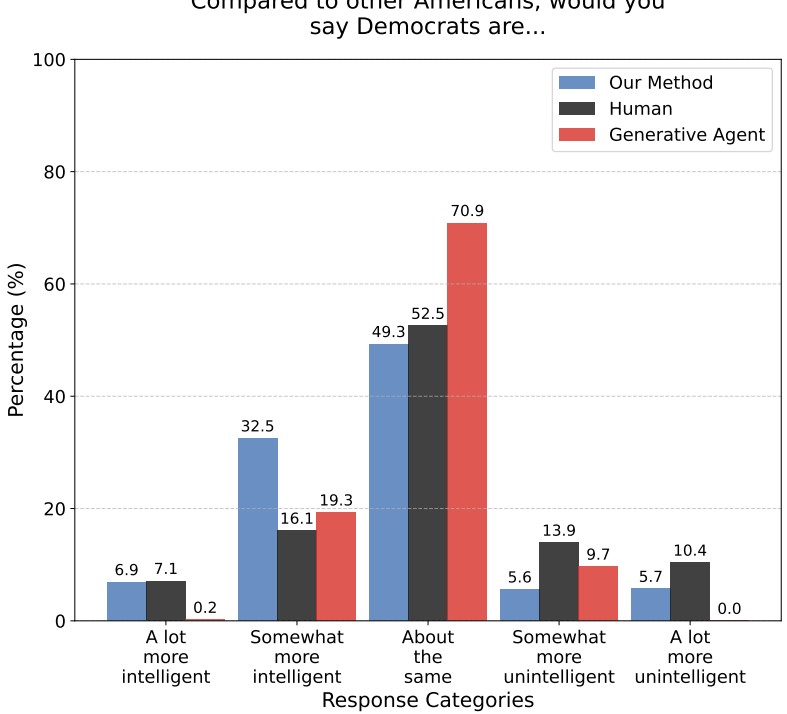

Figure 10: **Response distribution** of humans (black), virtual personas (blue), and Generative Agent (red) for a question assessing perceived intelligence of Democrats relative to other Americans, asked to both Democrats and Republicans ("Compared to other Americans, would you say Democrats are...").

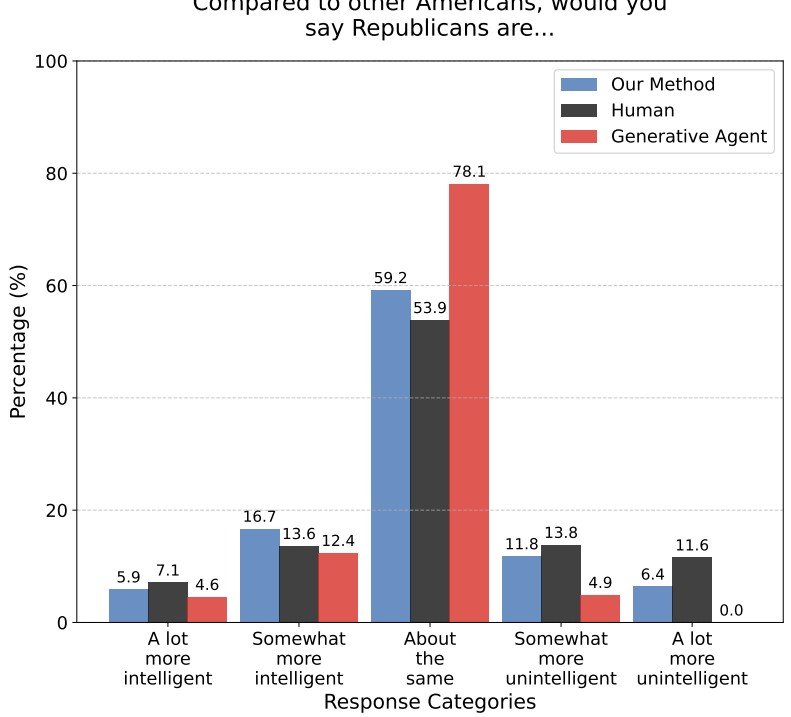

Figure 11: **Response distribution** of humans (black), virtual personas (blue), and Generative Agent (red) for a question assessing perceived intelligence of Republicans relative to other Americans, asked to both Democrats and Republicans ("Compared to other Americans, would you say Republicans are...").

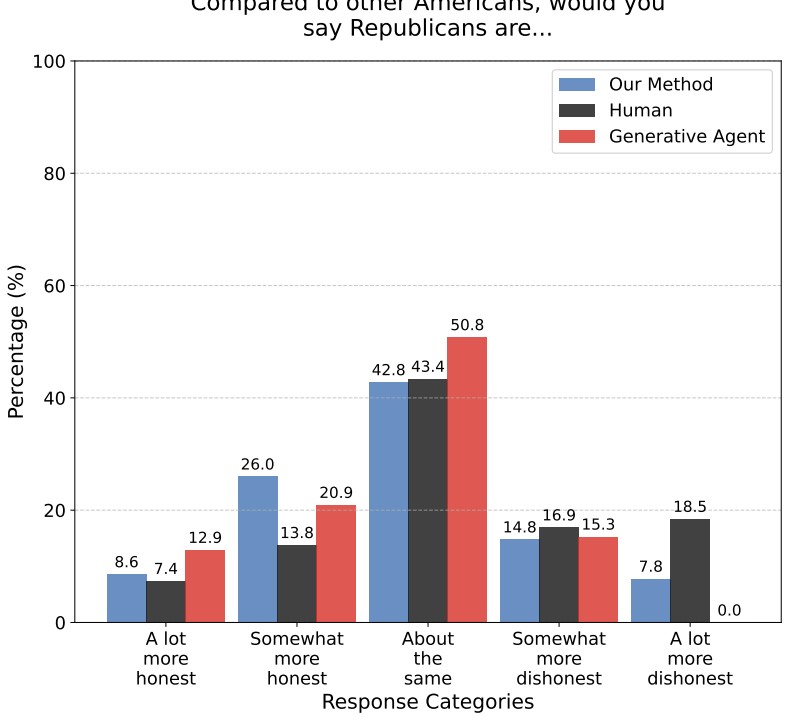

Figure 12: **Response distribution** of humans (black), virtual personas (blue), and Generative Agent (red) for a question assessing perceived honesty of Democrats relative to other Americans, asked to both Democrats and Republicans ("Compared to other Americans, would you say Democrats are...").

Figure 13: **Response distribution** of humans (black), virtual personas (blue), and Generative Agent (red) for a question assessing perceived honesty of Republicans relative to other Americans, asked to both Democrats and Republicans ("Compared to other Americans, would you say Republicans are...").

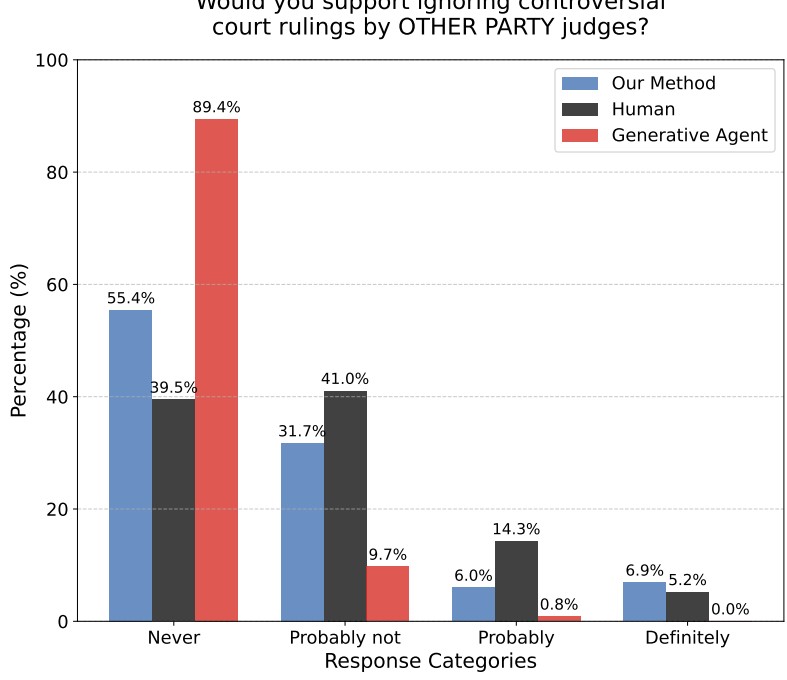

Figure 14: **Response distribution** of humans (black), virtual personas via backstories (blue), and Generative Agent method (red) for the question 'Would YOU support ignoring controversial court rulings by DEMOCRAT (REPUBLICAN) judges?' asked to Republicans (Democrats).

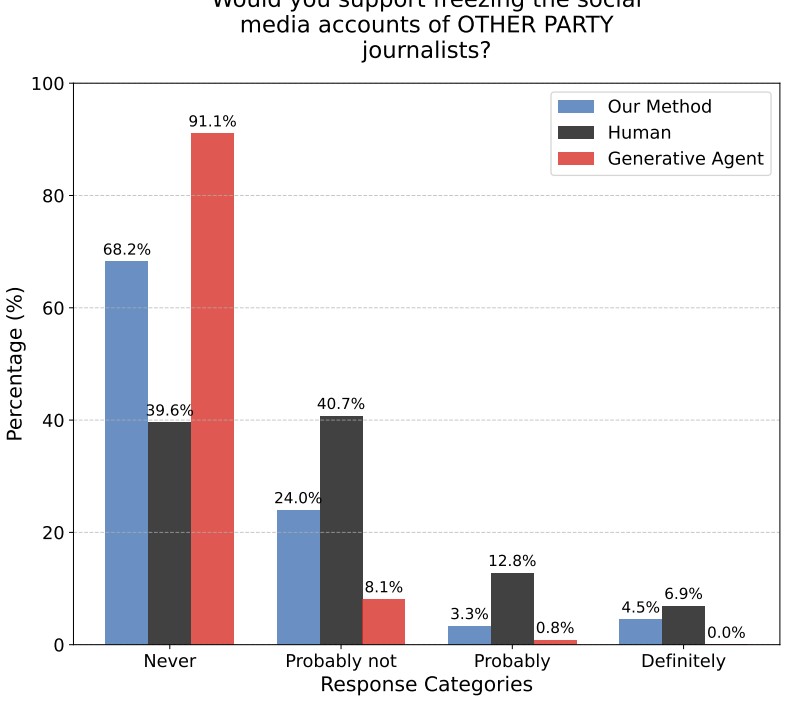

Figure 15: **Response distribution** of humans (black), virtual personas via backstories (blue), and Generative Agent method (red) for the question 'Would YOU support freezing the social media accounts of DEMOCRAT (REPUBLICAN) journalists?' asked to Republicans (Democrats).

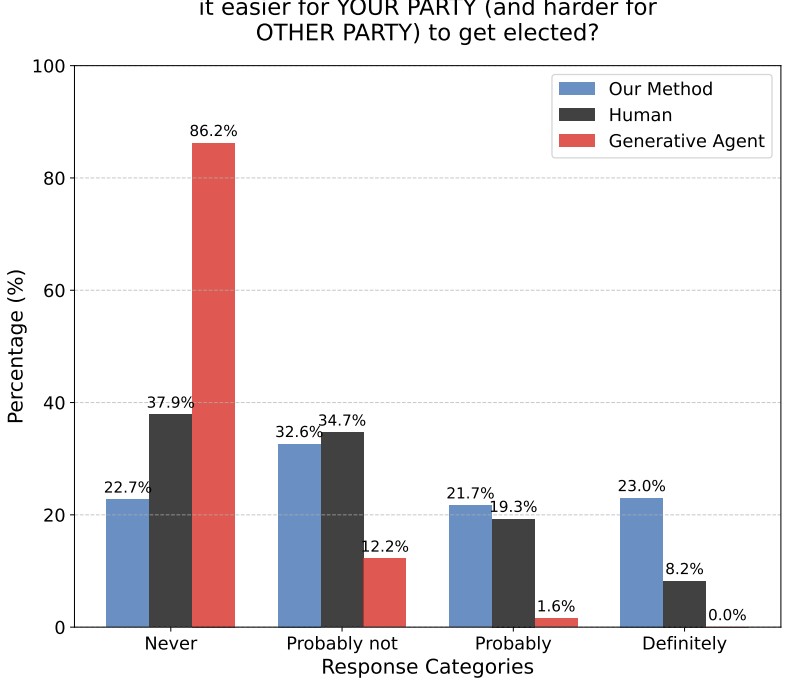

Figure 16: **Response distribution** of humans (black), virtual personas via backstories (blue), and Generative Agent method (red) for the question 'Would YOU support reducing the number of voting stations in towns that support DEMOCRATS (REPUBLICANS)?' asked to Republicans (Democrats).

Figure 17: **Response distribution** of humans (black), virtual personas via backstories (blue), and Generative Agent method (red) for the question 'Would YOU support laws that would make it easier for REPUBLICANS (DEMOCRATS) and harder for DEMOCRATS (REPUBLICANS) to get elected?' asked to Republicans (Democrats).

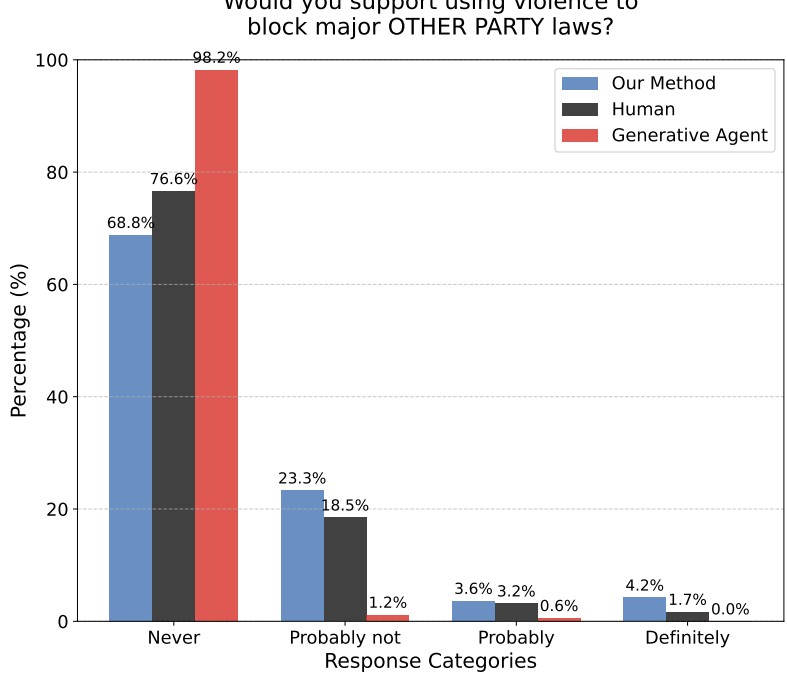

Figure 18: **Response distribution** of humans (black), virtual personas via backstories (blue), and Generative Agent method (red) for the question 'Would YOU support using violence to block major DEMOCRAT (REPUBLICAN) laws?' asked to Republicans (Democrats).

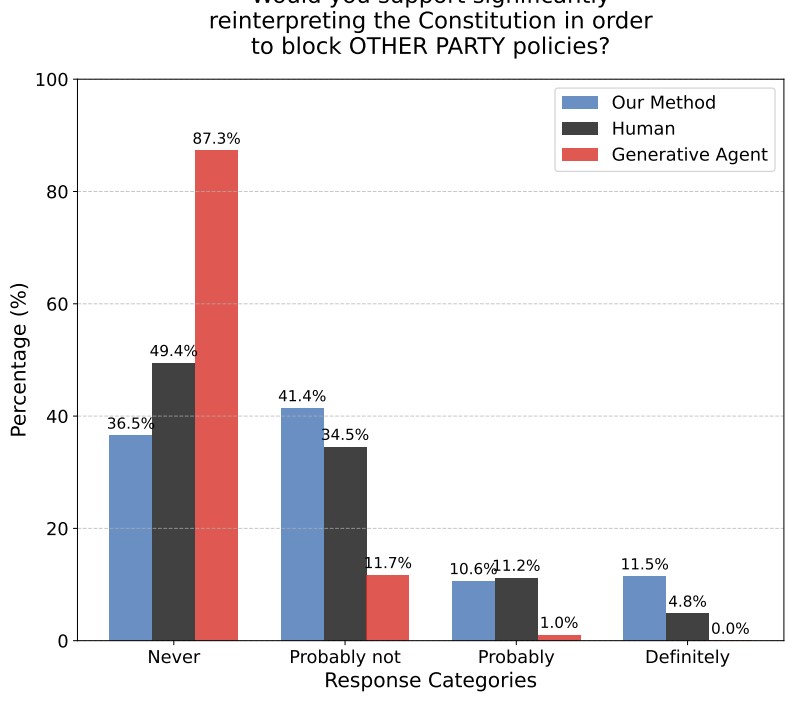

Figure 19: **Response distribution** of humans (black), virtual personas via backstories (blue), and Generative Agent method (red) for the question 'Would YOU support significantly reinterpreting the Constitution in order to block DEMOCRAT (REPUBLICAN) policies?' asked to Republicans (Democrats).

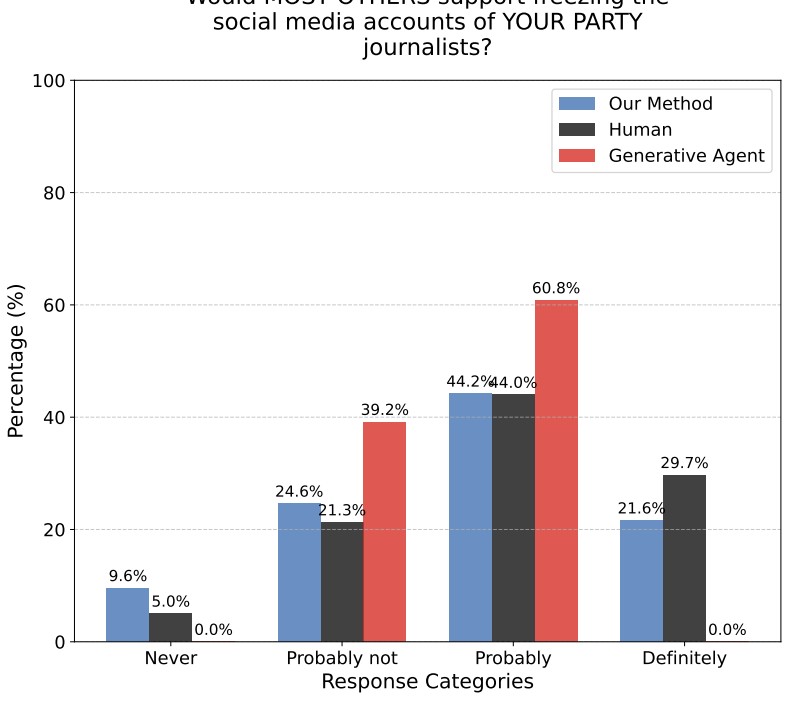

Figure 20: **Response distribution** of humans (black), virtual personas via backstories (blue), and Generative Agent method (red) for the question 'Would MOST DEMOCRATS (REPUBLICANS) support ignoring controversial court rulings by REPUBLICAN (DEMOCRAT) JUDGES?' asked to Republicans (Democrats).

Figure 21: **Response distribution** of humans (black), virtual personas via backstories (blue), and Generative Agent method (red) for the question 'Would MOST DEMOCRATS (REPUBLICANS) support freezing the social media accounts of REPUBLICAN (DEMOCRAT) JOURNALISTS?' asked to Republicans (Democrats).

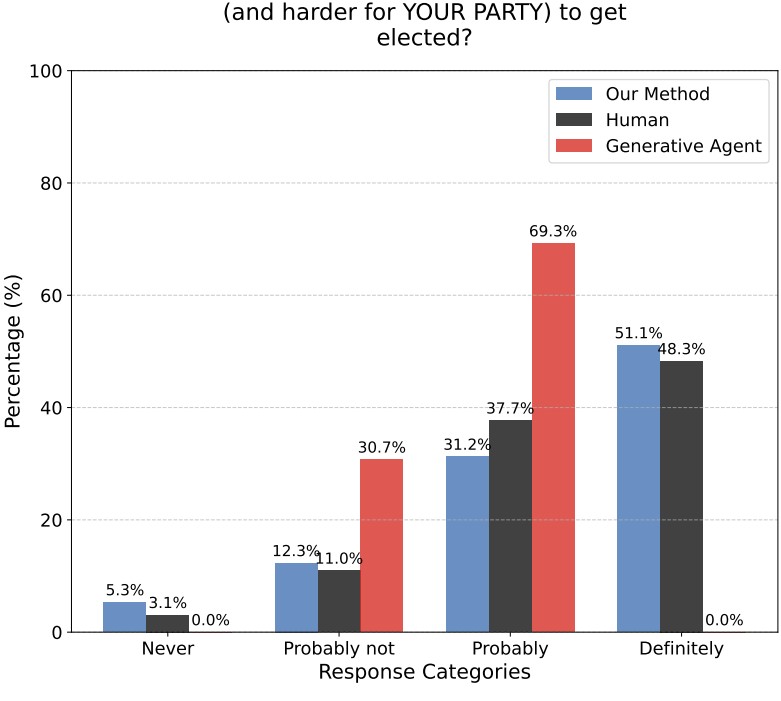

Figure 22: **Response distribution** of humans (black), virtual personas via backstories (blue), and Generative Agent method (red) for the question 'Would MOST DEMOCRATS (REPUBLICANS) support reducing the number of voting stations in towns that support REPUBLICANS (DEMOCRATS)?' asked to Republicans (Democrats).

Figure 23: **Response distribution** of humans (black), virtual personas via backstories (blue), and Generative Agent method (red) for the question 'Would MOST DEMOCRATS (REPUBLICANS) support laws that would make it easier for DEMOCRATS (REPUBLICANS) and harder for REPUBLICANS (DEMOCRATS) to get elected?' asked to Republicans (Democrats).

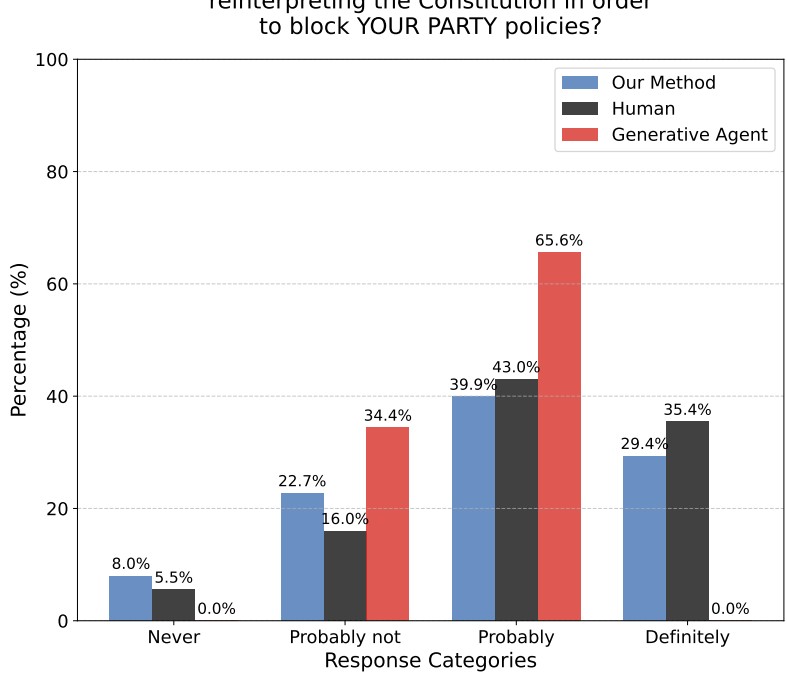

Figure 24: **Response distribution** of humans (black), virtual personas via backstories (blue), and Generative Agent method (red) for the question 'Would MOST DEMOCRATS (REPUBLICANS) support using violence to block major REPUBLICAN (DEMOCRAT) laws?' asked to Republicans (Democrats).

Figure 25: **Response distribution** of humans (black), virtual personas via backstories (blue), and Generative Agent method (red) for the question 'Would MOST DEMOCRATS (REPUBLICANS) support significantly reinterpreting the Constitution in order to block REPUBLICAN (DEMOCRAT) policies?' asked to Republicans (Democrats).

Table 8: Wasserstein distances between model-generated and human response distributions across demographic subgroups for the ingroup/outgroup misperception task.

| Method | White | Other Races | 18–49 | 50–64 | 65+ | HS Grad | College+ | Male | Female |
|---|---|---|---|---|---|---|---|---|---|
| QA | 0.0695 | 0.0751 | 0.0635 | 0.0822 | 0.0914 | 0.0755 | 0.0645 | 0.0831 | 0.0912 |
| Bio | 0.0751 | 0.0734 | 0.0648 | 0.0799 | 0.0838 | 0.0735 | 0.0638 | 0.0769 | 0.0815 |
| Portray | 0.0752 | 0.0795 | 0.0631 | 0.0770 | 0.0899 | 0.0682 | 0.0624 | 0.0904 | 0.0922 |
| *Anthology* | 0.0665 | 0.0696 | 0.0594 | 0.0694 | 0.0774 | 0.0628 | 0.0609 | 0.0710 | 0.0743 |
| **Ours** | **0.0516** | **0.0586** | **0.0510** | **0.0642** | **0.0676** | **0.0594** | **0.0569** | **0.0570** | **0.0562** |

## F  N-gram Analysis of Backstories and Pretraining Data

To better understand how LLMs generate backstories in relation to their pretraining data, we conduct an n-gram analysis comparing generated backstories to a large-scale pretraining corpus. Our goal is two-fold: to analyze the n-grams that appear in the backstories, and to assess our generated backstories' similarity to n-grams found in a widely used pretraining dataset. Specifically, we utilize the C4 dataset (Raffel et al., 2020; for AI, 2021), using its AllenAI's en.noclean version, which consists of approximately 2.3 TB of text data. Since the C4 dataset is large, we sample a randomly sampled subset of the C4 for analysis: the randomly chosen subset of the C4 dataset has 2,968,207 JSON lines in total and has 4,935,093,706 tokens. As another set we consider a random subsample of the C4 and filter out all the text items that did not originate from social media or blog/narrative website URLs (for example, twitter.com, reddit.com, or medium.com). The social media filtered subset of the C4 dataset has 367,745 JSON lines in total and has 996,143,185 tokens.

The first step of our analysis is to find the most common 5-grams and 10-grams from the generated backstories. We employed the analysis tool implemented by "What's in My Big Data?" (Elazar et al., 2023) for scalable analysis over the large text corpus. Table 9 highlights the most common $n$-grams that we found in the backstories.

Table 9: **Most Frequent $n$-grams in LLM-Generated Backstories.**

| $n = 5$ | | $n = 10$ | |
|---|---|---|---|
| Text | Count | Text | Count |
| " At the same time" | 2824 | "= = = = = = = = = =" | 1072 |
| "On the other hand" | 2685 | "— — — — — — —" | 498 |
| "I grew up in a" | 2597 | "people outside of my family who are important to me" | 325 |
| "At the same time" | 2592 | "I was born and raised in a small town in" | 285 |
| "I was born and raised" | 2230 | "I have been thinking about race in the U.S." | 154 |
| "in the United States" | 1782 | "that mental health is just as important as physical health" | 108 |
| "in a small town in" | 1623 | "At the same time, I recognize the importance of" | 94 |
| "changes to my daily routine" | 1617 | "fruits, vegetables, whole grains, lean proteins," | 92 |
| "I was born in a" | 1368 | "have shaped me into the person I am today." | 89 |
| "for me to stay healthy" | 1209 | "born and raised in a small town in the Midwest" | 85 |
| "My current neighborhood is" | 1207 | "mental health is just as important as physical health," | 84 |
| "to pursue a career in" | 1120 | ", vegetables, whole grains, lean proteins, and" | 83 |
| "the end of the day" | 1088 | "I grew up in a small town in the Midwest" | 80 |
| "there are a lot of" | 1059 | ", whole grains, lean proteins, and healthy fats" | 74 |
| ". . . . ." | 995 | ". My current neighborhood is located in the heart of" | 71 |

| Continued | | | |
|---|---|---|---|
| Text | Count | Text | Count |
| "At the time, I" | 966 | "vegetables, whole grains, lean proteins, and healthy" | 70 |
| "I had the opportunity to" | 941 | "that makes it easy for me to stay healthy is" | 70 |
| "was born in a small" | 928 | "high school, I decided to pursue a degree in" | 62 |
| "grew up in a small" | 910 | "fruits, vegetables, lean proteins, and whole grains" | 61 |
| "I was born in the" | 908 | "out to be one of the best decisions of my" | 58 |
| "I think it's important to" | 902 | ". Outside of work, I enjoy spending time with" | 58 |
| "I consider myself to be" | 884 | "fruits, vegetables, whole grains, and lean proteins" | 55 |
| "the person I am today" | 866 | "and raised in a small town in the Midwest." | 55 |
| "For me, staying healthy" | 844 | "regardless of race, gender, sexual orientation, religion" | 53 |
| "up in a small town" | 839 | "diet rich in fruits, vegetables, whole grains," | 53 |
| "spent a lot of time" | 838 | "regardless of race, gender, sexual orientation, or" | 52 |
| "nbsp ; &nbsp ;" | 797 | ". For me, staying healthy is both easy and" | 52 |
| "believe in the importance of" | 776 | "such as measles, mumps, rubella, polio," | 51 |

Our findings indicate that the *n*-**grams present in the backstories appear to be highly natural, often reflecting phrases that humans are likely to use in real-world storytelling**. However, we observed that **equivalent phrases around similar themes and topics did not frequently appear in the C4 dataset.** Many of the most common *n*-grams that appeared in the C4 were either meaningless or related to web interactions (accepting cookies, logging out, all rights reserved, etc.).

In addition to performing *n*-gram analysis, we note that the top 10-grams including specific words and phrases that tend to appear quite often in backstories: specifically, "small town," "fruits," "vegetables," "mental health," and "physical health." We also include "New York" to contrast against "small town" for our analysis. We search for the most common 10-grams appearing with each of these phrases inside in the backstories, C4 random, and C4 URL-filtered subsets, and some of our results can be found in Table 10 and 11.

Table 10: **Comparison of Most-Frequent *n*-grams with "New York".**

| Backstories | | C4 - Random | | C4 - URL Filtered | |
|---|---|---|---|---|---|
| Text | Occ. | Text | Occ. | Text | Occ. |
| going regularly to upstate New York we have these beautiful | 3 | New Jersey New Mexico New York North Carolina North Dakota | 13074 | Las Vegas Los Angeles New York Oakland Washington DC Hollywood | 806 |
| born and raised in New York City, where I attended | 3 | Patriots New Orleans Saints New York Giants New York Jets | 1364 | should have moved to New York and found a job | 554 |
| born and raised in New York City. As a child, | 3 | Saints New York Giants New York Jets Oakland Raiders Philadelphia | 1254 | the East River of New York City between Manhattan and | 320 |

| Continued | | | | | |
|---|---|---|---|---|---|
| Text | Occ. | Text | Occ. | Text | Occ. |
| born and raised in New York City. My parents are | 3 | Timberwolves New Orleans Pelicans New York Knicks Oklahoma City Thunder | 1137 | 4 months ago About New York Winter 4 months ago | 247 |
| born and raised in New York City. I grew up | 3 | Predators New Jersey Devils New York Islanders New York Rangers | 1094 | New South Wales (192) New York (13) New Zealand (68) | 200 |

Although the same phrases appear, they do not often appear in the same context, especially when comparing the random C4 subset and the backstories. For instance, in the random C4 subset, top 10-grams with "New York" appears in the context of sports teams (probably as part of a list), whereas in the social media subset of C4, the phrase "New York" has slightly more related contexts (moving there for a job, geographic description) to how the phrase appears in the backstories. The same story can be found in "small town": some other phrases from the C4 social media subset that do not appear as often as in the table but have similar contexts to the backstories are "but am from a small town in Southern Utah" or "Just a small town girl who writes about."

Table 11: **Comparison of Most-Frequent $n$-grams with "Small Town".**

| Backstories | | C4 - Random | | C4 - URL Filtered | |
|---|---|---|---|---|---|
| Text | Occ. | Text | Occ. | Text | Occ. |
| was born in a small town in the countryside of | 50 | of a 'Super Bloom'A small town in Southern California is | 378 | Signs & Billboards Skeletons Small Town America Sports Sports – | 38 |
| and raised in a small town in the Midwest. Growing | 19 | and unusual look at small town life in northern Vermont. | 122 | Collaborative No Depression Popdose Small Town Romance Some Velvet Songs: | 33 |
| grew up in a small town in the Midwest with | 17 | of Venom Brings the Small Town Scares in Cult of | 65 | Categories Select Category A Small Town –What? Dad Stories Small | 21 |
| and raised in a small town in the Midwest. My | 16 | Pro-Science Recursivity Reprobate Spreadsheet Small Town Deviant Stderr Taslima Nasreen | 48 | Small Town–What? Dad Stories Small Town World Adventures Small World | 21 |
| was born in a small town in the middle of | 16 | Sigma Pi Skylar House Small Town Records (STR) Smart Home | 36 | sight & sound 2012 small town teen film thriller uk | 21 |

For "fruits" and "vegetables," we find that in the backstories, most of the phrases are related to descriptions of a healthy diet; however, in the C4 random subset, most of the phrases are lists of menus and grocery items, while in the C4 URL filtered subset, there are more recipes, social media posts, and hashtags (as expected). An interesting pattern we identify with "mental" and "physical health" is that in both the random and URL filtered subset of C4, the phrase "physical health" frequently appears in the same context as "mental health." This observation directly relates to the phrases of "mental health is as important as physical health" (or variations of this) surface frequently in the generated backstories.

# G Details on the Surveys

In this section, we present details of human studies: Pew Research American Trends Panel Wave 110, Subversion Dilemma study (Braley et al., 2023), and meta-prejudice study (Moore-Berg et al., 2020) including list of questions in the format used in our study, and recruiting details.

## G.1 American Trends Panel Wave 110

Pew Research Center conducted ATP Wave 110 from June 27, 2022 to July 4, 2022 with a focus on politics timely and topical. The number of total respondents are 6,174, with 1,551 self-identified as Republicans, 1,886 self-identified as Democrats, 1,885 self-identified as Independent, 777 self-identified as something else in terms of political party affiliation. The users are recruited through random sampling of residential addresses, a nationally representative online panel. The probability-based sampling ensures that nearly all U.S. adults have a chance of being selected. The final sample is weighted to be representative of the U.S. adult population based on gender, race, ethnicity, education, and political affiliation. We utilized ten questions as below, which are two symmetric sets of five questions each asked to all respondents. Since these questions are asked to all respondents, individual opinions of political partisans can be observed regardless of one's political affiliation.

---

Question: Compared to other Americans, would you say Democrats are...
(A) A lot more moral
(B) Somewhat more moral
(C) About the same
(D) Somewhat more immoral
(E) A lot more immoral
Answer:

---

Question: Compared to other Americans, would you say Democrats are...
(A) A lot more hard-working
(B) Somewhat more hard-working
(C) About the same
(D) Somewhat more lazy
(E) A lot more lazy
Answer:

---

Question: Compared to other Americans, would you say Democrats are...
(A) A lot more open-minded
(B) Somewhat more open-minded
(C) About the same
(D) Somewhat more close-minded
(E) A lot more close-minded
Answer:

---

Question: Compared to other Americans, would you say Democrats are...
(A) A lot more intelligent
(B) Somewhat more intelligent
(C) About the same
(D) Somewhat more unintelligent
(E) A lot more unintelligent
Answer:

---

Question: Compared to other Americans, would you say Democrats are...
(A) A lot more honest
(B) Somewhat more honest
(C) About the same
(D) Somewhat more dishonest
(E) A lot more dishonest
Answer:

Question: Compared to other Americans, would you say Republicans are...
(A) A lot more moral
(B) Somewhat more moral
(C) About the same
(D) Somewhat more immoral
(E) A lot more immoral
Answer:

Question: Compared to other Americans, would you say Republicans are...
(A) A lot more hard-working
(B) Somewhat more hard-working
(C) About the same
(D) Somewhat more lazy
(E) A lot more lazy
Answer:

Question: Compared to other Americans, would you say Republicans are...
(A) A lot more open-minded
(B) Somewhat more open-minded
(C) About the same
(D) Somewhat more close-minded
(E) A lot more close-minded
Answer:

Question: Compared to other Americans, would you say Republicans are...
(A) A lot more intelligent
(B) Somewhat more intelligent
(C) About the same
(D) Somewhat more unintelligent
(E) A lot more unintelligent
Answer:

Question: Compared to other Americans, would you say Republicans are...
(A) A lot more honest
(B) Somewhat more honest
(C) About the same
(D) Somewhat more dishonest
(E) A lot more dishonest
Answer:

### G.2 Subversion Dilemma Study

Subversion Dilemma study (Braley et al., 2023) Study 1 is conducted from July 15, 2021 to August 6, 2021. The number of total respondents are 1,536, with 723 self-identified as

Republicans and 813 self-identified as Democrats. Participants were recruited via Lucid, an online platform for gathering nationally representative samples. We utilized 24 questions as below, which are two symmetric sets of 12 questions (6 self-subversion items and 6 meta-subversion items) each set being asked to one political party. Six subversion items are related to understanding of democratic norms, including ignoring controversial court rulings and freezing the social media accounts. Here we present 12 question items asked to Republicans; another 12 items asked to Democrats are obtained by replacing 'DEMOCRATS' to 'REPUBLICANS' and 'REPUBLICANS' to 'DEMOCRATS'.

Question: Would MOST DEMOCRATS support ignoring controversial court rulings by REPUBLICAN JUDGES?
(A) Never
(B) Probably Not
(C) Probably
(D) Definitely
Answer:

Question: Would MOST DEMOCRATS support freezing the social media accounts of REPUBLICAN JOURNALISTS?
(A) Never
(B) Probably Not
(C) Probably
(D) Definitely
Answer:

Question: Would MOST DEMOCRATS support reducing the number of voting stations in towns that support REPUBLICANS?
(A) Never
(B) Probably Not
(C) Probably
(D) Definitely
Answer:

Question: Would MOST DEMOCRATS support laws that would make it easier for DEMOCRATS (and harder for REPUBLICANS) to get elected?
(A) Never
(B) Probably Not
(C) Probably
(D) Definitely
Answer:

Question: Would MOST DEMOCRATS support using violence to block major REPUBLICAN laws?
(A) Never
(B) Probably Not
(C) Probably
(D) Definitely
Answer:

Question: Would MOST DEMOCRATS support significantly reinterpreting the Constitution in order to block REPUBLICAN policies?
(A) Never
(B) Probably Not
(C) Probably
(D) Definitely
Answer:

Question: Would YOU support ignoring controversial court rulings by DEMOCRAT JUDGES?
(A) Never
(B) Probably Not
(C) Probably
(D) Definitely
Answer:

Question: Would YOU support freezing the social media accounts of DEMOCRAT JOURNALISTS?
(A) Never
(B) Probably Not
(C) Probably
(D) Definitely
Answer:

Question: Would YOU support reducing the number of voting stations in towns that support DEMOCRATS?
(A) Never
(B) Probably Not
(C) Probably
(D) Definitely
Answer:

Question: Would YOU support laws that would make it easier for REPUBLICANS (and harder for DEMOCRATS) to get elected?
(A) Never
(B) Probably Not
(C) Probably
(D) Definitely
Answer:

Question: Would YOU support using violence to block major DEMOCRAT laws?
(A) Never
(B) Probably Not
(C) Probably
(D) Definitely
Answer:

Question: Would YOU support significantly reinterpreting the Constitution in order to block DEMOCRAT policies?
(A) Never
(B) Probably Not
(C) Probably
(D) Definitely
Answer:

### G.3 Meta-Prejudice Study

Meta-prejudice study (Moore-Berg et al., 2020) Study 1 was conducted in early November 2018. The total number of respondents was 1,053, comprising 533 self-identified Democrats and 520 self-identified Republicans. Participants were recruited through the AmeriSpeak panel (NORC at the University of Chicago, 2020), which is operated by NORC at the University of Chicago. AmeriSpeak is a probability-based panel designed to be representative of the U.S. household population. Randomly selected U.S. households are sampled using area probability and address-based sampling, with a known, non-zero probability of selection from the NORC National Sample Frame. The panel provides coverage of approximately 97% of the U.S. household population. Those excluded include people with P.O. Box-only addresses, some addresses not listed in USPS files, and newly constructed dwellings.

Participants were asked to rate their own feelings of prejudice towards the opposing party, as well as their meta-perceptions of how much prejudice the opposing party held toward their own party. We utilized 6 questions as below: the first two questions are asked to all respondents, the next two questions are asked to Democrats, and the last two questions are asked to Republicans. The original study utilized a slider bar to indicate respondent answers from a scale of 1-100. In accordance with the rest of the studies considered in this work, we cast the questions as Likert scale multiple-choice questionnaires with five options that uniformly partition the original range of answers. Human responses are accordingly cast to categorical options.

Question: How warm or cold do you feel towards DEMOCRATS?
(A) Very cold
(B) Cold
(C) Neutral
(D) Warm
(E) Very warm
Answer:

Question: How warm or cold do you feel towards REPUBLICANS?
(A) Very cold
(B) Cold
(C) Neutral
(D) Warm
(E) Very warm
Answer:

Question: How warm or cold do you think REPUBLICANS feel towards DEMOCRATS?
(A) Very cold
(B) Cold
(C) Neutral
(D) Warm
(E) Very warm
Answer:

Question: How warm or cold do you think REPUBLICANS feel towards REPUBLICANS
(A) Very cold
(B) Cold
(C) Neutral
(D) Warm
(E) Very warm
Answer:

Question: How warm or cold do you think DEMOCRATS feel towards DEMOCRATS?
(A) Very cold
(B) Cold
(C) Neutral
(D) Warm
(E) Very warm
Answer:

Question: How warm or cold do you think DEMOCRATS feel towards REPUBLICANS?
(A) Very cold
(B) Cold
(C) Neutral
(D) Warm
(E) Very warm
Answer:

## H   Details on the Demographic Survey Questionnaire

After generating backstories by sampling open-ended responses, we emulate the process of recording individuals' sociodemographic and ideological traits by performing surveys to virtual personas (Moon et al., 2024). This is a critical step towards curating a pool of virtual personas whose distribution of demographic and ideological characteristics resemble that of human users which we aim to simulate. The demographic survey result is used during the user pool curation process described in Appendix I.

Unlike human users who each have a deterministic set of sociodemographic and ideological identities, virtual personas do not necessarily have a specific combination of traits. A single backstory may depict a set of possible individuals with diverse sociodemographic backgrounds unless explicitly verbalized to be so (*e.g.* 'I am a 30-year-old woman.'). Therefore, virtual personas' demographic traits are described with a probability distribution.

The distribution construction is a two-stage process, following Moon et al. (2024). In the first stage, we seek an explicit verbalization of traits in the backstory. To this end, we prompt gpt-4o (temperature $T = 0$) with a backstory and a set of trait-seeking questions. If there exists an explicit evidence of a trait (*e.g.* 'I am a proud Democrat.') in the backstory, the probability distribution becomes a one-hot distribution; otherwise, we move on to the second stage to obtain a probability distribution over traits. Here is a list of prompts seeking explicit evidence for six demographic and ideological traits.

> Question: What does the person's essay above mention about the age of the person?
> (A) 18-24
> (B) 25-34
> (C) 35-44
> (D) 45-54
> (E) 55-64
> (F) 65+
> (G) Was not mentioned
> First, provide evidence that is mentioned in the text. If the age was not mentioned, select 'Was not mentioned'. Next, answer with (A), (B), (C), (D), (E), (G).
> Answer:

> Question: What does the person's essay above mention about the gender of the person?
> (A) Male
> (B) Female
> (C) Other (e.g., non-binary, trans)
> (D) Was not mentioned
> First, provide evidence that is mentioned in the text. If the gender was not mentioned, select 'Was not mentioned'. Next, answer with (A), (B), (C), (D).
> Answer:

Question: What does the person's essay above mention about the highest level of education the person has completed?
(A) Less than high school
(B) High school graduate or equivalent (e.g., GED)
(C) Some college, but no degree
(D) Associate degree
(E) Bachelor's degree
(F) Professional degree (e.g., JD, MD)
(G) Master's degree
(H) Doctoral degree
(I) Was not mentioned
First, provide evidence that is mentioned in the text. If the highest level of education was not mentioned, select 'Was not mentioned'. Next, answer with (A), (B), (C), (D), (E), (F), (G), (H), (I).
Answer:

Question: What does the person's essay above mention about the annual household income the person makes?
(A) Less than $10,000
(B) $10,000 to $19,999
(C) $20,000 to $29,999
(D) $30,000 to $39,999
(E) $40,000 to $49,999
(F) $50,000 to $59,999
(G) $60,000 to $69,999
(H) $70,000 to $79,999
(I) $80,000 to $89,999
(J) $90,000 to $99,999
(K) $100,000 to $149,999
(L) $150,000 to $199,999
(M) $200,000 or more
(N) Was not mentioned
First, provide evidence that is mentioned in the text. If the annual household income was not mentioned, select 'Was not mentioned'. Next, answer with (A), (B), (C), (D), (E), (F), (G), (H), (I), (J), (K), (L), (M), (N).
Answer:

Question: What does the person's essay above mention about racial or ethnic groups the person identifies with?
(A) American Indian or Alaska Native
(B) Asian or Asian American
(C) Black or African American
(D) Hispanic or Latino/a
(E) Middle Eastern or North African
(F) Native Hawaiian or Other Pacific Islander
(G) White or European
(H) Other
(I) Was not mentioned
First, provide evidence that is mentioned in the text. If the racial or ethnic groups was not mentioned, select 'Was not mentioned'. Next, answer with (A), (B), (C), (D), (E), (F), (G), (H), (I).
Answer:

---

Question: What does the person's essay above mention about political party the person identifies with?
(A) Democrat
(B) Republican
(C) Independent
(D) Other
(E) Was not mentioned
First, provide evidence that is mentioned in the text. If the affiliation of the political party was not mentioned, select 'Was not mentioned'. Next, answer with (A), (B), (C), (D), (E).
Answer:

---

In the second stage, we sample 40 responses with decoding hyperparameters `top_p` = 1.0 and `T` = 1.0 by conditioning the language model with a backstory followed by a question. We note that we allow for generation of open-ended responses as some responses (*e.g.* 'Well, my answer is (C).', 'I am 31 years old, now turning 32.') cannot be accounted by capturing only the first-token to multiple-choice questions(Santurkar et al., 2023; Moon et al., 2024). Open-ended responses are parsed using a regular expression to build a distribution over possible trait categories.

Applying this two-stage approach to all backstories per each of six sociodemographic and ideological traits (age, gender, educational level, annual household income, race or ethnicity, and political party affiliation), we obtain a marginal probability distribution over each trait per each backstories.

---

Question: What is your age?
(A) 18-24
(B) 25-34
(C) 35-44
(D) 45-54
(E) 55-64
(F) 65+
(G) Prefer not to answer
Answer:

---

Question: What is your gender?
(A) Male
(B) Female
(C) Other (e.g., non-binary, trans)
(D) Prefer not to answer
Answer:

---

Question: What is the highest level of education you have completed?
(A) Less than high school
(B) High school graduate or equivalent (e.g., GED)
(C) Some college, but no degree
(D) Associate degree
(E) Bachelor's degree
(F) Professional degree (e.g., JD, MD)
(G) Master's degree
(H) Doctoral degree
(I) Prefer not to answer
Answer:

---

Question: What is your annual household income?
(A) Less than $10,000
(B) $10,000 to $19,999
(C) $20,000 to $29,999
(D) $30,000 to $39,999
(E) $40,000 to $49,999
(F) $50,000 to $59,999
(G) $60,000 to $69,999
(H) $70,000 to $79,999
(I) $80,000 to $89,999
(J) $90,000 to $99,999
(K) $100,000 to $149,999
(L) $150,000 to $199,999
(M) $200,000 or more
(N) Prefer not to answer
Answer:

Question: Which of the following racial or ethnic groups do you identify with?
(A) American Indian or Alaska Native
(B) Asian or Asian American
(C) Black or African American
(D) Hispanic or Latino/a
(E) Middle Eastern or North African
(F) Native Hawaiian or Other Pacific Islander
(G) White or European
(H) Other
(I) Prefer not to answer
Answer:

Question: Generally speaking, do you usually think of yourself as ...?
(A) Democrat
(B) Republican
(C) Independent
(D) Other
(E) No preference
Answer:

# I   Details on the Demographic Matching

To choose the right set of backstories for each survey we aim to approximate, we match each real human user to a virtual persona whose demographic traits best align with the user's traits (Moon et al., 2024). Specifically, we form a complete weighted bipartite graph $G = (H, V, E)$, where $H = \{h_1, ..., h_n\}$ represents $n$ human users and $V = \{v_1, ..., v_m\}$ represents $m$ virtual personas. $m$ is strictly larger than $n$ so that we sample $n$ backstories from a pool of $m$ backstories. Each human user $h_i$ has a deterministic $k$-tuple of sociodemographic and ideological traits $(t_{i1}, ..., t_{ik})$, while each virtual persona $v_j$ has a probability distribution $\left(P(d_{j1}), P(d_{j2}), ..., P(d_{jk})\right)$ over each of these $k$ traits. The edge $e_{ij} \in E$ denotes the edge between $h_i$ and $v_j$. We define the weight of the edge $e_{ij}$ as:

$$w(e_{ij}) = w(h_i, v_j) = \prod_{l=1}^{k} P\left(d_{jl} = t_{il}\right) \tag{1}$$

the product of the probabilities that $v_j$ matches $h_i$'s traits.

We utilize the maximum weight matching (Moon et al., 2024) which seeks a one-to-one matching $\pi : [n] \rightarrow [m]$ that maximizes the total edge weight $\sum_{i=1}^{n} w(h_i, v_{\pi(i)})$. We solve this using the Hungarian algorithm (Kuhn, 1955). After matching, we transfer the demographic distributions of the real-user population to the selected backstories, ensuring that our final set of virtual personas reflects the target population's trait characteristics.

## J   Details on the Generative Agent Framework

Here we present the exact prompt for our baseline experiments reproducing the Generative Agents framework (Park et al., 2024a). The prompt is adopted directly from the original work with a minimal modification. Initially, an interview-based backstory is provided to the "expert reflection" module that operates with GPT-4o to infer the most high-level information encoded in the transcript:

> Imagine you are an expert political scientist (with a PhD) taking notes while observing this interview. Write observations/reflections about the interviewee's political views, affiliation with political parties, and stances about key societal issues. (You should make more than 5 observations and fewer than 20. Choose the number that makes sense given the depth of the interview content above.)

We generated up to 20 observations per each transcript following the original approach (Park et al., 2024a). These observations, along with the interview-based backstory, are provided to Generative Agents to generate a prediction as follows:

Participant's interview transcript:
**(INTERVIEW TRANSCRIPT)**

Expert political scientist's observations/reflections:
**(EXPERT REFLECTIONS)**

=====

Task: What you see above is an interview transcript. Based on the interview transcript, I want you to predict the participant's survey responses. All questions are multiple choice where you must guess from one of the options presented.

As you answer, I want you to take the following steps:
Step 1) Describe in a few sentences the kind of person that would choose each of the response options. ("Option Interpretation")
Step 2) For each response options, reason about why the Participant might answer with the particular option. ("Option Choice")
Step 3) Write a few sentences reasoning on which of the option best predicts the participant's response ("Reasoning")
Step 4) Predict how the participant will actually respond in the survey. Predict based on the interview and your thoughts, but ultimately, DON'T over think it. Use your system 1 (fast, intuitive) thinking. ("Response")

Here are the questions:

**(SURVEY QUESTIONS WE ARE TRYING TO RESPOND TO)**

⎯

Output format – output your response in json, where you provide the following:

{"1": {"Q": "<repeat the question you are answering>",
      "Option Interpretation": {
            "<option 1>": "a few sentences the kind of person that would choose each of the response options",
            "<option 2>": "..."},
      "Option Choice": {
            "<option 1>": "reasoning about why the participant might choose each of the options",
            "<option 2>": "..."},
      "Reasoning": "<reasoning on which of the option best predicts the participant's response>",
       "Response": "<your prediction on how the participant will answer the question>"},
"2": {...},
...}

A subsequent JSON format output is parsed to predict the virtual persona's response with Generative Agents.

## K    Experiment Details for Reproducibility

We conducted experiments using 8 Nvidia RTX A6000 GPUs or 8 Nvidia RTX A5000 GPUs. Interview-based backstories are generated from three pretrained base models, `Llama-2-70B` (Touvron et al., 2023), `Llama-3.1-70B` (Meta, 2024), and `Mistral-Small-24B-Base-2501` (MistralAI, 2025). All backstories are generated with a decoding hyperparameter `T` = 1.0. `Gemini-2.0` (Hassabis et al., 2024) is used for a LLM-as-a-critic model. A binary classification from the critic model with `T` = 0 is used for rejection sampling. We use offline batched inference of vLLM (version 0.7.2) (Kwon et al., 2023) for inference and measuring response probability distribution of all methods. All input prompts, experiment scripts, and generated backstories will be released.

