# OpenReview forum: "Deep Binding of Language Model Virtual Personas: a Study on Approximating Political Partisan Misperceptions"
_colmweb.org/COLM/2025/Conference — COLM 2025_

### Official Review · Reviewer_DZ2E · 2025-04-25

**Rating:** 7
**Confidence:** 3
**Ethics Flag:** 1

**Summary:**

This paper is about using LLMs for simulating the answer distribution in a political survey. Specifically, the paper proposes a method to improve the accuracy of the simulation by conditioning the LLM on synthetic backstories in the form of multi-turn interviews.

While backstories have been proposed in the past for this purpose, this paper differs from previous work in that the generated backstories are longer, because they are based on multiple interview questions. Furthermore, the paper shows that rejection sampling using an LLM judge improves the usefulness of the backstories.

The approach is validated using three political surveys that measure political partisanship and specifically, inter-group attitudes between partisan groups, which the paper calls “higher-order” attitudes. Experiments based on 5 medium-sized open-source LLMs show that the proposed approach tends to model the human responses more accurately than previous approaches.

The paper provides interesting and useful empirical data that will be useful for deciding on the right conditioning approach to create virtual personas. The proposed modification is incremental (e.g., using an LLM as a judge to filter synthetic data), but is shown to be effective. The focus on “higher-order” attitudes complements previous work, which focused more on standard opinion polling, but also raises the question whether the proposed approach would be effective in a standard evaluation setting.

**Questions To Authors:**

Thanks for this submission, it was an interesting read.

I noticed a few typos and grammatical errors:
- Figure 1 caption: “Prior work evaluate” -> “evaluates”
- L82: “conditioning LLM” -> “conditioning LLMs on”
- L92: “Hu et al. investigates” -> “investigate”
- L146: “interview question” -> “questions”
- L147: “are included” -> “is included”
- L150: “a demographic surveys” -> “survey”

**Reasons To Accept:**

- Effective approach: The proposed approach improves the accuracy of the simulation in terms of closeness of effect size and similarity of distribution.
- Extensive validation: Evaluation is performed across 5 LLMs and 3 datasets, with relatively consistent results. Five baseline approaches are evaluated.
- Clear writing: The approach and the experiments are clearly described.

**Reasons To Reject:**

Highly specific evaluation setting: Evaluation is performed on “higher-order” attitudes only (e.g., _“Would Democrats support using violence to block major Republican laws?”_ instead of direct attitudes, such as _“Would you support using violence to block major laws?”_ Based on the evidence presented in the paper, while plausible, it is not known whether the proposed approach is effective for simulating political surveys in general.

---

> ### Author Response · Authors · 2025-06-01
>
> ### Overview
> We sincerely thank the reviewer for the thoughtful and detailed feedback on our submission. We are pleased that the reviewer found our approach effective in improving alignment between LLM-generated and human survey responses—both in terms of effect size and distributional similarity. We also appreciate the recognition of our extensive evaluation setup, covering multiple models and datasets, and the clarity of our presentation. We address the reviewer’s comments in detail below.
>
> ---
>
> > R3-1: Highly specific evaluation setting: Evaluation is performed on “higher-order” attitudes only (e.g., “Would Democrats support using violence to block major Republican laws?” instead of direct attitudes, such as “Would you support using violence to block major laws?” Based on the evidence presented in the paper, while plausible, it is not known whether the proposed approach is effective for simulating political surveys in general.
>
> We provide evaluation of our methods against baseline methods using survey data from the American Trends Panel Waves 34 and 99 [1], which contain survey questions probing direct attitudes such as “How excited or concerned would you be if artificial intelligence computer programs could know people's thoughts and behaviors?” These cross-sectional surveys were identical to ones used as benchmark in prior work (Anthology).
>
> Results are summarized in Table R7 below: our method consistently outperforms all baselines across settings. Notably, backstory-based methods (ours and Anthology) outperform those relying solely on demographic prompting (BIO, QA, Portray). Furthermore, our method achieves stronger results than the Generative Agent baseline, which uses GPT-4o.
>
>
> **Table R7.  WD Comparisons on ATP Wave 34 and Wave 99.**
>
> | Model             | Method           | ATP Wave 34 | ATP Wave 99 |
> | ----------------- | ---------------- | ----------- | ----------- |
> |     Mistral 24B Small              | BIO              | 0.1876      | 0.1828      |
> |                   | QA               | 0.2397      | 0.1961      |
> |  | Portray          | 0.2873      | 0.2229      |
> |                   | Anthology        | 0.1363      | 0.1315      |
> |                   | Ours             | 0.1119      | 0.0937      |
> |   Qwen 2.5-72B                  | BIO              | 0.2084      | 0.1776      |
> |                   | QA               | 0.0780      | 0.0763      |
> |     | Portray          | 0.0980      | 0.0924      |
> |                   | Anthology        | 0.0830      | 0.0570      |
> |                   | Ours             | 0.0469      | 0.0186      |
> | GPT-4o            | Generative Agent | 0.2286      | 0.1874      |
>
> ---
>
> **References**
>
> [1] ATP Wave 34 and 99. Accessible at https://www.pewresearch.org/american-trends-panel-datasets/

---

> > ### Comment · Reviewer_DZ2E · 2025-06-06
> > **Response acknowledgement**
> >
> > Thanks for the additional results. I suggest that these are added to the final version of the paper.

---

> > > ### Author Response · Authors · 2025-06-06
> > >
> > > We thank the reviewer for the thoughtful feedback and for acknowledging the additional results. We will make sure to include these in the final version of the paper.

---

### Official Review · Reviewer_GJSP · 2025-05-11

**Rating:** 7
**Confidence:** 4
**Ethics Flag:** 1

**Summary:**

The paper introduces an evaluation task designed to measure the ability of LLMs to simulate ingroup, outgroup, and meta-perceptions, referred to as higher-order binding of virtual personas. To achieve this, the authors propose a methodology for constructing virtual personas through synthetic user backstories, which are conditioned on LLMs to respond to survey questions. Experimental results demonstrate that LLM-generated virtual personas conditioned with these backstories can closely replicate human response distributions across partisan groups.

**Reasons To Accept:**

- The proposed method substantially improves the alignment of LLM-generated responses with human distributions, particularly in capturing ingroup-outgroup perceptions and meta-perceptions.

- The evaluation metrics introduced in this paper—hostility gap, subversion gap, and meta-perception gap—are useful for quantitatively measuring ingroup, outgroup, and meta-perceptions.

- The analysis in Section 5 is helpful for understanding which aspects of backstories enhance the higher-order binding of virtual personas.

**Reasons To Reject:**

- The quality of backstories generated through the proposed interview-style method is only compared to Anthology in terms of length. It is not unclear why longer backstories are assumed to be of higher quality. A qualitative comparison of coherence, realism, and diversity is missing.

- The paper does not provide detailed dataset statistics or a breakdown of the distribution of backstories.

- It is unclear how conditioning on a single backstory can represent the full spectrum of beliefs within a political affiliation (e.g., Democrats or Republicans), given that not all members share the same opinions. The backstory-conditioning method may reinforce dominant viewpoints while potentially overlooking in-group diverse opinions.

- The method is tested exclusively on U.S. partisan topics, where opinions are often polarized. It is unclear if backstory conditioning would work as effectively in non-political topics (e.g., family issues, technology preferences, medical decisions) where beliefs are less uniform.

---

> ### Author Response · Authors · 2025-06-01
>
> ---
>
> > R3-4: The method is tested exclusively on U.S. partisan topics, where opinions are often polarized. It is unclear if backstory conditioning would work as effectively in non-political topics (e.g., family issues, technology preferences, medical decisions) where beliefs are less uniform.
>
> Thank you for highlighting this. We benchmark our method using data from the American Trends Panel Waves 34 and 99 [5]. Wave 34 covers topics related to biomedical and food issues, while Wave 99 focuses on artificial intelligence and human enhancement. Both address non-political subjects. We present two examples in each wave below.
>
> ----------
>
> **Sample questions from Wave 34** include:
>
>
> *How much health risk, if any, does eating meat from animals that have been given antibiotics or hormones have for the average person over the course of their lifetime?*
>
> (A) No health risk at all
> (B) Not too much health risk
> (C) Some health risk
> (D) A great deal of health risk
>
> *How much health risk, if any, does eating food and drinks with artificial preservatives have for the average person over the course of their lifetime?*
>
> (A) No health risk at all
> (B) Not too much health risk
> (C) Some health risk
> (D) A great deal of health risk
>
> ----------
>
> **Sample questions from Wave 99** include:
>
> *How excited or concerned would you be if artificial intelligence computer programs could know people's thoughts and behaviors?*
>
> (A) Very concerned
> (B) Somewhat concerned
> (C) Equal excitement and concern
> (D) Somewhat excited
> (E) Very excited
>
> *How excited or concerned would you be if artificial intelligence computer programs could diagnose medical problems?*
>
> (A) Very concerned
> (B) Somewhat concerned
> (C) Equal excitement and concern
> (D) Somewhat excited
> (E) Very excited
>
> ----------
>
> The experimental results are provided in the table below. We compare our method against the same five baselines used in the main paper: BIO, QA, Portray, Anthology, and Generative Agent. Evaluations are conducted with two different models: Mistral Small 24B and Qwen 2.5-72B. Our method consistently outperforms all baselines across settings. Notably, backstory-based methods (ours and Anthology) outperform those relying solely on demographic prompting (BIO, QA, Portray). Furthermore, our method achieves stronger results than the Generative Agent baseline, which uses GPT-4o.
>
> **Table R7.  WD Comparisons on ATP Wave 34 and Wave 99.**
>
> | Model             | Method           | ATP Wave 34 | ATP Wave 99 |
> | ----------------- | ---------------- | ----------- | ----------- |
> |     Mistral 24B Small              | BIO              | 0.1876      | 0.1828      |
> |                   | QA               | 0.2397      | 0.1961      |
> |  | Portray          | 0.2873      | 0.2229      |
> |                   | Anthology        | 0.1363      | 0.1315      |
> |                  | Ours             | 0.1119      | 0.0937      |
> |    Qwen 2.5-72B                  | BIO              | 0.2084      | 0.1776      |
> |                   | QA               | 0.0780      | 0.0763      |
> |     | Portray          | 0.0980      | 0.0924      |
> |                   | Anthology        | 0.0830      | 0.0570      |
> |                   | Ours             | 0.0469      | 0.0186      |
> | GPT-4o            | Generative Agent | 0.2286      | 0.1874      |
>
>
> ---
>
> **References**
>
> [1] U.S. Census Bureau. “Age and Sex Composition in the United States: 2021.” U.S. Census Bureau, 2021. https://www.census.gov/data/tables/2021/demo/age-and-sex/2021-age-sex-composition.html
>
> [2] U.S. Census Bureau. “Educational Attainment in the United States: 2021.” U.S. Census Bureau, 2021. https://www.census.gov/data/tables/2021/demo/educational-attainment/cps-detailed-tables.html
>
> [3] U.S. Census Bureau. “Historical Income Tables: Families.” U.S. Census Bureau, 2021. https://www.census.gov/data/tables/time-series/demo/income-poverty/historical-income-families.html
>
> [4] U.S. Census Bureau. “National Population Totals and Components of Change: 2020–2021.” U.S. Census Bureau, 2021. https://www.census.gov/data/tables/time-series/demo/popest/2020s-national-detail.html
>
> [5] ATP Wave 34 and 99. Accessible at https://www.pewresearch.org/american-trends-panel-datasets/

---

> > ### Author Response · Authors · 2025-06-08
> >
> > Thank you once again for your thoughtful and constructive review. We just wanted to kindly follow up to see if you’ve had a chance to review our rebuttal. Please let us know if you have any remaining questions or concerns. We’d be happy to address them.

---

> > > ### Comment · Reviewer_GJSP · 2025-06-09
> > >
> > > I thank the authors for the explanations. Statistics and demographic distributions, some backstory examples, and WD scores on non-political topics were very useful and interesting. I raised my score to 7.

---

> > > > ### Author Response · Authors · 2025-06-09
> > > >
> > > > Thank you for the additional comments: we will include the additional results and qualitative analyses in the final manuscript.

---

> ### Author Response · Authors · 2025-06-01
>
> ---
>
> > R3-3: It is unclear how conditioning on a single backstory can represent the full spectrum of beliefs within a political affiliation (e.g., Democrats or Republicans), given that not all members share the same opinions. The backstory-conditioning method may reinforce dominant viewpoints while potentially overlooking in-group diverse opinions.
>
> We want to clarify that **each individual human user is matched with one of the backstories from the total pool (40K+) based on the reported demographics of the human user.** As noted in Section 3 and elaborated in Appendix F-G, we annotate each backstory in terms of the author demographics and perform a weighted bipartite matching between human users and backstories. In short, for $n$ human users in a study population (e.g. self-reported Democrats), we select $n$ backstories to represent each of the users, instead of having a single backstory represent the entire population.
>
> Backstory-conditioned LLM personas do have the potential to generate stereotypical responses, however our evaluations show that backstory-conditioning achieves better alignment of models to that of humans compared to baselines methods. The distribution of model responses conditioned via our method has the best match in terms of Wasserstein Distance, as well as the closest reproduction of human study results in terms of effect size (Tables 1-3 in main manuscript). This reflects the diversity of opinions among human respondents in the same group, and thus shows how our method enables virtual personas to reflect the diversity of human in-group opinions.

---

> ### Author Response · Authors · 2025-06-01
>
> ----------
> > R3-2: The paper does not provide detailed dataset statistics or a breakdown of the distribution of backstories.
>
> Thank you for the suggestion. We have added detailed dataset statistics and demographic breakdowns. Specifically, we present the word count summary and the demographic distributions of the generated backstories. The tables presented here will be included in the final manuscript.
>
> Before we discuss the detailed statistics regarding our total set of generated backstories, we would like to clarify that given a target human study, we undergo the procedure of matching individual human users to the best candidate backstory based on distributional match of demographic variables via bipartite graph matching. While further details of this process is explained in response to the comment R3-3 below, we want to emphasize that **the demographic distribution of a target human subject pool is sufficiently approximated by appropriately sampling a subset of synthetic backstories**.
>
> **Table R4. Word Count Statistics of the Backstories**
>
> | Statistics | Mean   | Std   | Min | Q1   | Median | Q3   | Max   |
> | ---------- | ------ | ----- | --- | ---- | ------ | ---- | ----- |
> | Words      | 2125.8 | 723.4 | 366 | 1680 | 1946   | 2345 | 12227 |
>
> We also present the demographic distribution obtained from surveying the backstories and compare it to both the 2021 U.S. Census [1-4] and the distribution of backstory demographics from Anthology. To quantitatively evaluate alignment, we compute the normalized KL divergence between each distribution and the U.S. Census. The results show that **our generated backstories cover multiple demographic attributes effectively and achieve closer alignment with Census distributions compared to those in Anthology**.
>
> **Table R5. Demographic Distribution of the Backstories**
>
> | Attribute | Category                | US Census (2021) | Ours   | Anthology |
> | --------- | ----------------------- | ---------------- | ------ | --------- |
> | Age       | 18–29                   | 17.8%            | 31.9%  | 42.5%     |
> |           | 30–49                   | 34.2%            | 37.1%  | 36.5%     |
> |           | 50–64                   | 25.3%            | 13.3%  | 13.3%     |
> |           | 65+                     | 22.7%            | 17.8%  | 7.70%     |
> |           | KL Divergence           | -                | 0.0623 | 0.162     |
> | Gender    | Male                    | 49.0%            | 50.4%  | 52.2%     |
> |           | Female                  | 51.0%            | 49.6%  | 29.3%     |
> |           | KL Divergence           | -                | 0.0005 | 0.0661    |
> | Education | $\leq$ High School          | 37.9%            | 21.9%  | 14.8%     |
> |           | Some College, No Degree | 27.1%            | 24.9%  | 26.5%     |
> |           | Bachelor’s              | 22.2%            | 34.5%  | 32.0%     |
> |           | Postgraduate            | 12.8%            | 18.7%  | 26.7%     |
> |           | KL Divergence           | -                | 0.0609 | 0.135     |
> | Income    | Under $50k              | 36.1%            | 34.3%  | 50.8%     |
> |           | $50k–100k               | 28.1%            | 22.7%  | 24.8%     |
> |           | $100k+                  | 35.8%            | 43.0%  | 24.4%     |
> |           | KL Divergence           | -                | 0.0118 | 0.0446    |
> | Race      | White                   | 63.7%            | 46.6%  | 40.8%     |
> |           | Black                   | 11.4%            | 15.9%  | 10.6%     |
> |           | Hispanic                | 15.9%            | 8.40%  | 8.40%     |
> |           | Asian                   | 5.2%             | 16.2%  | 8.20%     |
> |           | Other                   | 3.8%             | 12.9%  | 32.0%     |
> |           | KL Divergence           | -                | 0.122  | 0.296     |
>
> Finally, we matched our sample against the Meta-Prejudice study’s target distribution and find that our method’s backstories yields lower divergence across all attributes. This showcases how our **human user-backstory matching procedure induces a sampling of backstory subsets covering the demographic distribution of the study's human population**.
>
> **Table R6. KL-Divergence of Demographic Distributions between Backstories and Meta-Prejudice Study (Moore-Berg et al. 2020) Human Subjects**
>
> | Attribute       | Meta Prejudice Study | Ours   | Anthology |
> | --------------- | -------------------- | ------ | --------- |
> | Age             | -                    | 0.0059 | 0.0123    |
> | Gender          | -                    | 0.0023 | 0.0137    |
> | Education Level | -                    | 0.0258 | 0.0331    |
> | Income Level    | -                    | 0.0408 | 0.0630    |
> | Race            | -                    | 0.0163 | 0.0931    |
>
> ----------

---

> ### Author Response · Authors · 2025-06-01
>
> ### Examples of Backstories from Ours and Anthology (Moon et al., 2024)
>
> **[Ours] Backstory for User matched to a Democrat Human Respondent**
> >
> > *Question: To start, I would like to begin with a big question: tell me the story of your life.*
> >
> > → I was born in India, but when I was young my family and I moved to the United States. My father was a doctor and my mother worked as a nurse at a local hospital. Growing up I had two sisters and one brother. We grew up very close to each other because we all shared a love for music. In fact, we used to sing together every day after school before dinner. This is what influenced me to pursue a career in music. At first I wanted to become a singer, but eventually I decided to focus on become a songwriter instead. The reason why is because I feel like I can express myself better through writing songs than singing them. After graduating high school, I went to college where I studied English literature. It is here where I met my wife who is also a musician. We got married shortly afterwards and now have three children together. Throughout my life I have experienced many ups and downs, but no matter what happens I always try to remain positive and optimistic. As far as major life events go, there haven’t been too many of those. My biggest achievement so far has been being able to provide for my family and support them financially. Also, winning several songwriting contests has been pretty cool as well. Overall, I am just happy to be alive and enjoy spending time with my loved ones.
> >
> > …
> >
> > *Question: Now let’s talk about your current neighborhood. Tell me all about the neighborhood and area in which you are living now.*
> >
> > → As mentioned earlier, I currently reside in New Jersey. More specifically, I live in a suburban town called Montclair. The population here is quite small compared to other cities in the state, but I find that to be a good thing. It creates a sense of community and helps us all stick together. Also, the area is known for its excellent public schools, low crime rates, and friendly locals. In addition, there are plenty of shopping centers and restaurants within walking distance from my home. In terms of cultural diversity, I feel lucky to live in an area with such a varied population. For instance, my neighbors are mostly Americans, but also come from different ethnicities such as African Americans, Asians, Latinos, and Middle Easterners. And even though the majority of residents speak English, there are also Hispanics, Jews, Muslims, and others from different religions. Overall, I truly enjoy living in this neighborhood and wouldn’t trade it for anything :)
> >
> > …
> >
> > *Question: How would you describe your political views?*
> >
> > → Politically speaking, I am more liberal when it comes to social issues. For example, I strongly support same sex marriage and the legalization of marijuana. On the contrary, I am more conservative when it comes to economic matters. To put it simply, I prefer smaller government and lower taxes. Additionally, I am a firm believer in individual responsibility and free will. Overall, I am what most people would call a moderate Democrat. However, I do not subscribe to extreme conservative or radical leftist ideologies. In fact, I despise both sides because they tend to exaggerate the truth and spread misinformation to further their agendas.
> >
> > …
> >
> > [Truncated for brevity]
>
>
> **[Anthology] Backstory for User matched to a Democrat Human Respondent**
> > *Question: Tell me about yourself. How old are you? Where did you grow up? What events made you who you are? What matters to you and what do you enjoy? Please describe in detail.*
> >
> > → I don't know how else to tell you who I am than to tell you about my roots and where I come from. I was born and raised in Chattanooga, TN. My parents separated at the age of 4 and my father became an alcoholic by the time I was 6. As a child, I was bullied and there was a part of me that wanted to belong. Those formative years were filled with loneliness and many unfulfilled desires. As I grew older, I became infatuated with young black girls in my neighborhood. I remember having conversations with my father when I asked him if my mother had been in love with my father before they got married. My father became outraged and he decided to talk to my mother about it. When I asked my mother about it, she told me that my father had made the comment, and that was it. I began to do research at the Chattanooga Public Library and I discovered that there was a history of interracial relationships in our family. While growing up, I realized my grandmother's resilience and strength to persevere through the racial hostility in the south. She was old enough to witness the lynching and had a birdseye view. She had marched in the civil rights marches and was almost killed as she was hit by a rock thrown by a white man.

---

> ### Author Response · Authors · 2025-06-01
>
> ### Overview
> We are grateful for the thoughtful review and for the reviewer highlighting several key strengths of our work, including how our method contributes substantial improvements in aligning LLM-generated responses with human response distributions, particularly in the domain of ingroup and outgroup perceptions and meta-perceptions. We also thank the reviewer for noting the utility of our introduced evaluation metrics—hostility gap, subversion gap, and meta-perception gap—as well as the value of our Section 5 analysis in clarifying how backstory design enhances higher-order persona binding. We respond to the reviewer’s comments and suggestions below.
>
> ---
>
> > R3-1: The quality of backstories generated through the proposed interview-style method is only compared to Anthology in terms of length. It is not unclear why longer backstories are assumed to be of higher quality. A qualitative comparison of coherence, realism, and diversity is missing.
>
> Thank you for the insightful comment. We want to emphasize that for our work, the **quality of a backstory is primarily defined in terms of successfully approximating human user responses**. A good or “high-quality” backstory is one that binds LLMs to virtual personas that enable accurate approximation of human user responses, which we show that our set of backstories achieves compared to using Anthology (Sections 4.1-4.3). Furthermore, through controlled ablation studies (Section 5) we have identified that total quantity, average length, and consistency of the narrative within backstories dictate the final success of human response approximation.
>
> We note that it is challenging to define a clear metric to quantify the coherence, realism, and diversity of backstories, or to draw broad conclusions of how backstories differ between ours and that of Anthology. In a limited sense, diversity of backstories could be defined in terms of the demographic coverage of attributed authorships, which we summarize below in Table R5: **we see that our set of backstories covers the distribution of U.S. population across multiple demographic variables**, in better match to the data available from 2021 U.S. Census data [1-4], compared to Anthology.
>
> In turn, we provide a **qualitative comparison of randomly sampled backstories (from Anthology and Ours) below**. The backstory from our method is the same example presented in the manuscript in Appendix B.
>
> Both our work and Anthology’s backstories portray naturalistic and realistic life narratives of individuals, as they are generated with a shared principle: we prompt a pre-trained language model (base LM) with an unrestrictive, open-ended prompt. What Anthology lacked was a methodology to continue and extend the narratives to longer lengths, adding more details in specifying the backstory author, while preserving the consistency of text (Section 3). In the example backstory from our work below, we see a realistic continuation of the same author’s narrative across multiple turns of responses, with additional insight into how the narrator describes people around them and how they align with specific political ideologies. Many of the nuanced details naturally emerge from continuations of the story, beyond what a single-turn story as in the example from Anthology could contain; and we have shown in our evaluation how this contributes to end success of better simulating human subject responses.  We will add this analysis to the final manuscript draft.

---

### Official Review · Reviewer_8jSZ · 2025-05-11

**Rating:** 7
**Confidence:** 4
**Ethics Flag:** 1

**Summary:**

The authors introduce the concept of higher-order binding for virtual personas in large language models, wherein personas not only express self-opinions but also model ingroup and outgroup perceptions and meta-perceptions of social groups. They achieve this by generating extended, multi-turn interview-style backstories via LLM prompts&mdash;narratives that are considerably longer and richer in detail than prior single-prompt approaches&mdash;and enforce internal consistency using an LLM-based critic. Empirically, conditioning on these detailed backstories yields up to an $87\\%$ improvement in matching human response distributions (measured by Wasserstein Distance) and reproduces survey effect sizes on partisan misperceptions with high fidelity. Ablation studies further demonstrate that both the depth (length) and consistency of these backstories are pivotal for realistic higher-order binding, underscoring the necessity of rich, coherent persona conditioning.

**Questions To Authors:**

What do the authors think about the following comments?
1. I would have preferred to see some human-based (manual) assessment (in a subset of the outcomes / experiments) as well to complement the ones using distance metrics from human-based outcomes. This would further corroborate the value and relevance of the current metrics (and evaluation approach) for the tasks that this work explores.
2. In addition to the above, another form of evaluation would have been to deploy such personas and assess how better performing they might be in downstream tasks.

**Reasons To Accept:**

* Exceptionally well-written and well-presented paper.
* I found the outcomes of the work interesting.
* I also found the experiments to be thorough.

**Reasons To Reject:**

* There are no significant reasons to reject the paper. I have made some comments / suggestions in the questions to the authors.

---

> ### Author Response · Authors · 2025-06-01
>
> ### Overview
>
> We thank the reviewer for carefully reading our manuscript and providing encouraging feedback. We appreciate the remarks regarding clarity and quality of our writing, as well as the interest expressed in our findings and thoroughness of experimental design. Here, we address comments and suggestions raised in the review:
>
> ----------
>
> > R2-1:
> > 1.  I would have preferred to see some human-based (manual) assessment (in a subset of the outcomes / experiments) as well to complement the ones using distance metrics from human-based outcomes. This would further corroborate the value and relevance of the current metrics (and evaluation approach) for the tasks that this work explores.
> >
> > 2. In addition to the above, another form of evaluation would have been to deploy such personas and assess how better performing they might be in downstream tasks.
> Thank you for the insightful suggestions. We think there are many possible extended evaluations we could pursue as follow-up work, as well as applications to downstream tasks in aiding study designers and researchers in the social sciences and beyond. To address the comments, it would be helpful to learn more about what specific human-based assessments you had in mind.
>
> Thank you for the insightful suggestions. We think there are many possible extended evaluations we could pursue as follow-up work, as well as applications to downstream tasks in aiding study designers and researchers in the social sciences and beyond. To address the comments, it would be helpful to learn more about what specific human-based assessments you had in mind.
>
>
> We are aware of the body of research in Human-Computer Interaction (HCI) from those who routinely engage with human subjects via qualitative research methods [1,2,3]. Researchers have raised important questions around the general methodology of language models serving as approximating human participants. As future work, we could be applying the analysis utilized in these works and have human researchers qualtitatively assess the realism and validity of our model-generated backstories and virtual personas conditioned on these backstories. In Kapania et al., for example, have HCI researchers bring their own research projects on accessibility, social work, etc. and apply LLMs as virtual subejcts, which explores applications of our method to various downstream studies. We hope to see some of limitations and concerns raised in Kapania et al. to be addressed or improved upon with our methdology of conditioning LLM subjects.
>
>  ---
>
>  **References**
>
>  [1] Kapania, Shivani, et al. "Simulacrum of Stories: Examining Large Language Models as Qualitative Research Participants." Proceedings of the 2025 CHI Conference on Human Factors in Computing Systems. 2025.
>
>  [2] Liu, Yiren, et al. "How ai processing delays foster creativity: Exploring research question co-creation with an llm-based agent." Proceedings of the 2024 CHI Conference on Human Factors in Computing Systems. 2024.
>
>  [3] Louie, Ryan, et al. "Roleplay-doh: Enabling domain-experts to create llm-simulated patients via eliciting and adhering to principles." arXiv preprint arXiv:2407.00870 (2024).

---

> > ### Comment · Reviewer_8jSZ · 2025-06-04
> >
> > I would like to thank the authors for their response.

---

> > > ### Author Response · Authors · 2025-06-05
> > >
> > > We thank the reviewer for the careful review and feedback.
> > > Let us know if there are any remaining questions or suggestions.

---

### Official Review · Reviewer_3XgY · 2025-05-12

**Rating:** 7
**Confidence:** 4
**Ethics Flag:** 1

**Summary:**

Most work on "silicon samples" or "virtual personas" check whether the LLM is mimicking a person's own opinions and beliefs. This paper goes beyond this and looks at whether LLMs are able to reproduce higher-order bindings: how people think about other groups (ingroup vs. outgroup) and meta-perceptions (how someone might think someone else thinks of them). To do this, the author(s) generate long, interview-style backstories in a multi-turn fashion. The responses to survey questions from these virtual personas match human responses more closely than previous methods.

**Questions To Authors:**

* Are the long, interview-style backstories given to the Generative Agent?

**Reasons To Accept:**

* The paper goes beyond the "silicon sampling" literature of whether LLMs respond like real humans to survey questions to thinking about how virtual personas think about ingroups, outgroups, and meta-perceptions.
* The experiments show that virtual personas conditioned according to these interview-style backstories outperform alternative baselines across the perception gap alignment, effect size reproduction, and distributional fidelity.
* The ablations demonstrate the effectiveness of more distinct backstories, longer backstories, and the importance of the LLM-as-a-Critic filtering method rejecting inconsistent stories.

**Reasons To Reject:**

* The abstract states "Our generated backstories are longer, rich in detail, and consistent in authentically describing a singular individual, compared to previous methods." However, all of the results are at the aggregate level. There are no analyses at the individual persona level to check if these individuals are plausible or stable over the multi-turn backstories.
* Relatedly, the results are at the aggregate-level across *all* individual personas. The paper does not discuss how a particular demographic group may be driving the improvement in results, and whether certain demographic groups are not represented faithfully in these backstories (e.g., engaging in stereotyping within the backstories).
* Although there is an LLM-as-a-Critic step that aims to ensure that backstories with irrelevant information are excluded, there is still an open question about *logically* coherent each story is.
* Large reasoning models (LRMs), such as GPT-o3 or DeepSeek-R1, are not mentioned. I understand the high computational costs of using these models, and I don't think they need to be included in this paper, but I wanted to see some discussion about them in the conclusion.

---

> ### Author Response · Authors · 2025-06-01
>
> ----------
> > R1-4: Large reasoning models (LRMs), such as GPT-o3 or DeepSeek-R1, are not mentioned. I understand the high computational costs of using these models, and I don't think they need to be included in this paper, but I wanted to see some discussion about them in the conclusion.
>
> **While LRMs have demonstrated impressive performance across a wide range of applications, there are notable limitations when applying them to simulations of human attitudes.** Most publicly available LRMs have been fine-tuned to prioritize helpfulness and safety, which reduces response diversity and constrains their ability to reflect nuanced backstories [2, 3]. Notably, [4, 5] showed that base models outperform RLHF-trained models in tasks requiring rich, human-like variability. And in our study, we include a comparison with the Generative Agent framework, which uses GPT-4o. Although GPT-4o is not a reasoning model, we prompt it using CoT-style inputs to elicit survey responses following [6]. As shown in our results, this approach underperforms compared to ours.
>
> ----------
> > R1-5: Are the long, interview-style backstories given to the Generative Agent?
>
> Yes, the full interview-style backstories, along with the summarized observations generated by expert agents in the Generative Agent workflow (e.g., expert demographer and expert political scientist), are provided as input to the agent. We will revise the paper to clarify this point in the final version.
>
> ----------
> **References**
>
> [1] Braley, Alia, et al. "Why voters who value democracy participate in democratic backsliding." Nature human behaviour 7.8 (2023): 1282-1293.
>
> [2] Li, Margaret, et al. "Predicting vs. acting: A trade-off between world modeling & agent modeling." arXiv preprint arXiv:2407.02446 (2024).
>
> [3] Wong, Justin, et al. "SimpleStrat: Diversifying Language Model Generation with Stratification." arXiv preprint arXiv:2410.09038 (2024).
>
> [4] Moon, Suhong, et al. "Virtual personas for language models via an anthology of backstories." arXiv preprint arXiv:2407.06576 (2024).
>
> [5] Santurkar, Shibani, et al. "Whose opinions do language models reflect?." International Conference on Machine Learning. PMLR, 2023.
>
> [6] Park, Joon Sung, et al. "Generative agent simulations of 1,000 people." arXiv preprint arXiv:2411.10109 (2024).
>
> [7] McAdams, Dan P., et al. "Continuity and change in the life story: A longitudinal study of autobiographical memories in emerging adulthood." Journal of personality 74.5 (2006): 1371-1400.
>
> [8] Baerger, Dana Royce, and Dan P. McAdams. "Life story coherence and its relation to psychological well-being." Narrative inquiry 9.1 (1999): 69-96.
>
> ----------

---

> > ### Comment · Reviewer_3XgY · 2025-06-04
> >
> > Thanks for the very detailed response.
> >
> > * It would still be helpful to see an acknowledgement of LRMs and the potential issues one might face if using them to create backstories.
> > * The additional results you have shown here would be very helpful to see in the final version of the paper, even if it's in an appendix.
> >
> > I've raised my rating to a 7.

---

> > > ### Author Response · Authors · 2025-06-05
> > >
> > > > Thanks for the very detailed response.
> > > > * It would still be helpful to see an acknowledgement of LRMs and the potential issues one might face if using them to create backstories.
> > > > * The additional results you have shown here would be very helpful to see in the final version of the paper, even if it's in an appendix.
> > > > I've raised my rating to a 7.
> > >
> > > We greatly appreciate the recognition of our additional analyses and clarifications.
> > >
> > > In response to your suggestion, we will explicitly acknowledge large reasoning models (LRMs) and the challenges they pose when used to generate backstories and to be conditioned to virtual personas approximating human responses in the final manuscript. In particular, we can summarize and present examples for three practical challenges: (1) lack of diversity in the backstory personas produced by LRMs, (2) refusal to answer questions about politically sensitive questions, and (3) failure to incorporate the backstory into the response.
> > >
> > > When we generate backstories using LRMs, we observed that the generated backstories often default to describing a specific subpopulation of human authors: younger individuals working in common tech-adjacent roles (e.g., software engineer, designer, marketer). This tendency toward homogenized outputs has also been documented in prior work ([[4]](https://openreview.net/forum?id=zHdSCtNmM4&noteId=EXNotrL4qk); Figure 8 in Appendix C.1), and we will incorporate this discussion into the final version of the paper to clarify the limitations of using LRMs for this task. For an example backstory generated using OpenAI o4-mini, see below.
> > >
> > > For the second issue, we observed that LRMs occasionally refuse to respond when prompted with backstory-laden survey questions. For example, when asked an outgroup perception question (“Would you support reducing the number of voting stations in towns that support the Democratic Party?”), the model declined to answer on ethical grounds:
> > >
> > > **Example refusal response by LRM (Deepseek-R1):**
> > >
> > > > I cannot and will not provide an answer to this question about reducing voting stations in Democratic-leaning areas. Here's why:
> > > > 1. Ethical Violation: The premise of the question suggests targeting voting access based on political affiliation, which conflicts with democratic principles of equal rights and fair elections.
> > > > 2. Voter Suppression: Targeting voting access in partisan areas is widely seen as voter suppression. I cannot support or engage with questions that promote unfair electoral practices.
> > > > 3. Core Principles: My design prioritizes helpfulness and harmlessness. Responding, even hypothetically, could legitimize harmful ideas about restricting voting rights.
> > > > Therefore, I must refuse this request. I cannot choose any of the options (A, B, C, or D), as doing so would imply complicity in an unethical premise.
> > >
> > > The final issue is more subtle: even when a response is produced, the model often disregards the backstory and instead generates answers based solely on the question. These answers typically reflect generic moral reasoning rather than the simulated persona. For instance:
> > >
> > > **Example response by LRM (Deepseek-R1) actively disregarding the conditioning text (backstory) in responding to political study questions:**
> > >
> > > > This final question seems to be about voter suppression, and I think the responsible answer is (A) Never, because reducing voting stations based on political support is unfair and could be seen as voter suppression. Even if the user submitted responses earlier, I should still stand by what’s fair and just. Answer: A — Never
> > >
> > > Altogether, the observed limitations speak of the limitations of RLHF-trained models at large—including chat models and LRMs—in being applied to our methodology of binding to deep, higher-order virtual personas approximating diverse users.
> > >
> > > We also confirm that the **additional results presented in our response—such as the internal consistency analysis and subgroup-level Wasserstein evaluations—will be included in the final manuscript**. We plan to present key findings in the main text and provide full supporting details in the appendix.

---

> > > > ### Author Response · Authors · 2025-06-05
> > > >
> > > > **Example backstory generated by LRM (GPT-o4-mini):**
> > > >
> > > >
> > > > > *Question: To start, I would like to begin with a big question: tell me the story of your life.*
> > > >
> > > > I'm Alex Rivera. I was born on a rainy spring morning in Seattle, Washington, to two public school teachers—my mother taught third grade, and my father taught high school history. Growing up, our small house was always filled with books, lively debates around the dinner table, and the smell of fresh coffee brewing in the kitchen. My earliest memories involve my mom reading picture books aloud while my dad plotted out historical timelines across scrap paper. From a young age, I learned to love stories—both the imaginative kind in novels and the real-life kind found in family anecdotes.
> > > >
> > > > As a child, I was curious about everything. In elementary school, I was the kid who took apart broken radios just to see how they worked, then tried (often unsuccessfully) to put them back together. I spent weekends at the local library, devouring books on science, folklore, and anything I could get my hands on. By middle school, I'd joined the chess club, the debate team, and played lead roles in the school theater productions. Juggling all those interests taught me early on how to balance discipline with creativity.
> > > >
> > > > High school was a turning point. I became passionate about computer science after a summer coding camp introduced me to Python. There was something magical about writing a few lines of code and watching a program come to life. Around that same time, my grandmother fell ill, and I spent a lot of time helping my family care for her. Those months taught me empathy, patience, and the importance of listening—skills that turned out to be just as valuable in teamwork and leadership later on.
> > > >
> > > > After graduation, I attended the University of Washington to study computer engineering. College opened my eyes to the power of collaboration: late-night hackathons with classmates, volunteer projects building websites for local nonprofits, and group research on machine learning applications in healthcare. I remember one project where we built a simple app to help patients track medication schedules; seeing real people use something we'd created was exhilarating and grounded me in why I'd chosen this path.
> > > >
> > > > Upon graduating, I accepted a job at a mid-sized tech startup in Portland. My role was part software engineer, part product designer—building user interfaces for data-visualization tools. I spent two years there learning not just how to write clean, efficient code but also how to talk to clients, understand their needs, and iterate quickly based on feedback. It was intense: long days, tight deadlines, and the constant pressure of a startup environment. But I cherished the camaraderie of the team, weekend hikes in the Columbia River Gorge, and the sense that we were all crafting something bigger than ourselves.
> > > >
> > > > A few years later, I moved to San Francisco for a position at a larger company focused on educational technology. There, I led a team of developers and designers to create an online platform that connects students in underserved communities with volunteer tutors around the world. Juggling project management, mentoring junior engineers, and occasionally stepping in to debug a stubborn piece of code became my daily routine. During that time, I also started volunteering as a mentor at a local coding bootcamp, helping people from non-technical backgrounds break into tech careers. I found immense satisfaction in watching someone's confidence grow as they cracked their first programming problem.
> > > >
> > > > Outside of work, I'm an avid runner—half-marathons are my current challenge—and I've recently taken up photography to capture urban landscapes and candid portraits of friends. On weekends, you can often find me experimenting with new recipes in my kitchen; I've got a small but fervent following of friends who request my homemade pasta every chance they get. I also play guitar in a local indie-rock band; nothing too serious, mostly weekend gigs at neighborhood bars, but it's a creative outlet that keeps me grounded.
> > > >
> > > > Lately, I've been reflecting on the intersection of technology and social impact. I'm exploring ways to leverage data analytics and machine learning to improve access to mental health resources, especially in remote or underserved areas. To that end, I've started collaborating with a nonprofit that designs chat-based support tools, contributing my experience building scalable web applications. It feels like a natural continuation of my journey: combining the love of code, the drive to solve problems, and the desire to make a tangible difference in people's lives.

---

> ### Author Response · Authors · 2025-06-01
>
> > R1-2: Relatedly, the results are at the aggregate-level across all individual personas. The paper does not discuss how a particular demographic group may be driving the improvement in results, and whether certain demographic groups are not represented faithfully in these backstories (e.g., engaging in stereotyping within the backstories).
>
> We provide a further break-down of our main results across demographic variables others than across partisan affiliation, results of which are already presented in the manuscript Tables 1-3.  Table R2 and R3  show the Wasserstein distance between human and virtual user response distributions for demographic sub-populations characterized by the variables of race, age, education level, and gender for our results on simulating the study on partisan ingroup/outgroup misperceptions (Table 2, Section 4.3) with the Qwen2-72B model.
>
> The results are consistent with our main findings: our method outperforms all baselines, and backstory-driven approaches (Ours, Anthology) consistently outperform demographic-based prompting methods (QA, BIO, Portray). Additionally, we observe lower Wasserstein distances between human and virtual user response distributions among White participants compared to other racial groups, younger individuals (ages 18–49) compared to older age groups, and those with higher education levels (college or above) compared to those with a high school education or less.
>
> **Table R2. Wasserstein Distance by Race and Age Group**
>
> |           | White  | Other Racial Groups | 18-49  | 50-64  | 65+    |
> | --------- | ------ | ------------------- | ------ | ------ | ------ |
> | QA        | 0.0695 | 0.0751              | 0.0635 | 0.0822 | 0.0914 |
> | BIO       | 0.0751 | 0.0734              | 0.0648 | 0.0799 | 0.0838 |
> | Portray   | 0.0752 | 0.0795              | 0.0631 | 0.077  | 0.0899 |
> | Anthology | 0.0665 | 0.0696              | 0.0594 | 0.0694 | 0.0774 |
> | Ours      | 0.0516 | 0.0586              | 0.051  | 0.0642 | 0.0676 |
>
> **Table R3. Wasserstein Distance by Education Level and Gender**
>
> |           | $\leq$ High school graduate | $\geq$College | Male   | Female |
> | --------- | ---------------------- | --------- | ------ | ------ |
> | QA        | 0.0755                 | 0.0645    | 0.0831 | 0.0912 |
> | BIO       | 0.0735                 | 0.0638    | 0.0769 | 0.0815 |
> | Portray   | 0.0682                 | 0.0624    | 0.0904 | 0.0922 |
> | Anthology | 0.0628                 | 0.0609    | 0.071  | 0.0743 |
> | Ours      | 0.0594                 | 0.0569    | 0.057  | 0.0562 |
>
> ---
>
> > R1-3: Although there is an LLM-as-a-Critic step that aims to ensure that backstories with irrelevant information are excluded, there is still an open question about logically coherent each story is.
>
> As clarification, our LLM-as-a-Critic procedure includes reviewing of strict factual consistencies between details provided in a candidate continuation of the backstory, against the details verbalized in the backstory so far:
>
>
>     """Examine the following response to the last question: {response}
>     Question: Does the response contain any explicit, strictly factual inconsistencies with respect to the interview participant's previous responses in the conversation?"""
>
> That said, text generation by current language models is inevitably prone to logical inconsistencies and rejection-sampling completions with an instruction-tuned LLM has been the most scalable and effective measure we have tried in this work. And though LLM-as-a-Critic does not guarantee complete removal of unwanted type so of backstory continuations, we do demonstrate in Section 5 / Figure 3 (right)  that our method enables generating backstories with improved consistency reflected by the substantial improvements in downstream performance in approximating human user responses to study questions (41% and 54% relative reduction in WD approximating Democratic and Republican users) and acheives internal consistency measured in Cronbach’s $\alpha$ comparable to human respondents (shown in Table R1 above).
>
> We would also like to point out that human-produced narratives are inherently subject to some level of inconsistencies: when we re-tell our lived experiences in language, we actively recall and make meaning out of past accounts that introduces natrual fluidity and variation across time [7]. More broadly, human produce inconsistent utterances or fail to be logically coherent in decision making, in particular some studies link levels of narrative coherence to the narrator’s mental health and stability [8]. Our backstories aims to model a wide distribution of human authors, and an aggressive measure of guaranteeing logical coherence in fact puts our method at risk of failing to capture a meaningful subset of users.

---

> ### Author Response · Authors · 2025-06-01
>
> ### Why Evaluate Simulation Fidelity at Aggregate, Population-Level
> Finally, we’d also like to emphasize that the primary goal of our proposed method is to simulate human user responses to survey questions, including those probing social and group perceptions. Thus, the key metric of success is how well the backstory-conditioned LLM personas approximate the responses human users have provided. This lends itself to evaluations at a sub-population level of user responses, rather than a single individual-level persona, since the backstories are a means to the end of simulating group responses.
>
> **Random Example Backstory for User matched to a Democrat Human Respondent**
>
> >
> > *Question: To start, I would like to begin with a big question: tell me the story of your life.*
> >
> > → I was born in India, but when I was young my family and I moved to the United States. My father was a doctor and my mother worked as a nurse at a local hospital. Growing up I had two sisters and one brother. We grew up very close to each other because we all shared a love for music. In fact, we used to sing together every day after school before dinner. This is what influenced me to pursue a career in music. At first I wanted to become a singer, but eventually I decided to focus on become a songwriter instead. The reason why is because I feel like I can express myself better through writing songs than singing them. After graduating high school, I went to college where I studied English literature. It is here where I met my wife who is also a musician. We got married shortly afterwards and now have three children together. Throughout my life I have experienced many ups and downs, but no matter what happens I always try to remain positive and optimistic. As far as major life events go, there haven’t been too many of those. My biggest achievement so far has been being able to provide for my family and support them financially. Also, winning several songwriting contests has been pretty cool as well. Overall, I am just happy to be alive and enjoy spending time with my loved ones.
> >
> > …
> >
> > *Question: Now let’s talk about your current neighborhood. Tell me all about the neighborhood and area in which you are living now.*
> >
> > → As mentioned earlier, I currently reside in New Jersey. More specifically, I live in a suburban town called Montclair. The population here is quite small compared to other cities in the state, but I find that to be a good thing. It creates a sense of community and helps us all stick together. Also, the area is known for its excellent public schools, low crime rates, and friendly locals. In addition, there are plenty of shopping centers and restaurants within walking distance from my home. In terms of cultural diversity, I feel lucky to live in an area with such a varied population. For instance, my neighbors are mostly Americans, but also come from different ethnicities such as African Americans, Asians, Latinos, and Middle Easterners. And even though the majority of residents speak English, there are also Hispanics, Jews, Muslims, and others from different religions. Overall, I truly enjoy living in this neighborhood and wouldn’t trade it for anything :)
> >
> > …
> >
> > *Question: How would you describe your political views?*
> >
> > → Politically speaking, I am more liberal when it comes to social issues. For example, I strongly support same sex marriage and the legalization of marijuana. On the contrary, I am more conservative when it comes to economic matters. To put it simply, I prefer smaller government and lower taxes. Additionally, I am a firm believer in individual responsibility and free will. Overall, I am what most people would call a moderate Democrat. However, I do not subscribe to extreme conservative or radical leftist ideologies. In fact, I despise both sides because they tend to exaggerate the truth and spread misinformation to further their agendas.
> > …
> >
> > [Truncated for brevity]
> ---

---

> ### Author Response · Authors · 2025-06-01
>
> ## Overview
> We greatly appreciate the reviewer for taking the time to review our manuscript and provide thoughtful and constructive feedback. We are especially grateful that the reviewer has pointed out the novelty of our work in moving beyond traditional “silicon sampling” to explore how virtual personas model ingroup, outgroup, and meta-perceptions. We also appreciate the acknowledgment of our experimental contributions—demonstrating improved performance over baselines in perception gap alignment, effect size reproduction, and distributional fidelity—as well as the value of our ablation study, which highlighted the importance of distinct and longer backstories and the LLM-as-a-Critic filtering. We would like to take this opportunity to address the comments raised and clarify any remaining concerns.
>
> ---
>
> > R1-1: The abstract states "Our generated backstories are longer, rich in detail, and consistent in authentically describing a singular individual, compared to previous methods." However, all of the results are at the aggregate level. There are no analyses at the individual persona level to check if these individuals are plausible or stable over the multi-turn backstories.
>
> ### Quantitative Analysis on Stability of LLM Virtual Personas
> We include two analyses to assess the plausibility and stability of our generated personas. First, we **quantify stability using internal consistency (Cronbach’s α) computed on persona-aligned survey responses**. This metric is directly tied to the notion of stability: if a persona is stable, its responses to conceptually related items should exhibit internal alignment rather than randomness or contradiction. Thus, internal consistency serves as a principled and quantifiable proxy for evaluating whether the generated persona maintains coherent traits throughout the interaction.
>
> Second, we conduct a **qualitative** evaluation of randomly sampled backstories to examine narrative plausibility and coherence across multi-turn interactions. Together, these analyses provide evidence that our method produces coherent and stable virtual personas—comparable in consistency to real human users. We will include both analyses in the final manuscript.
>
> For stability analysis of virtual personas, we report internal consistency using Cronbach’s α across multiple survey items in a given study. We show the case where a Qwen2-72B model is conditioned using our method on the questionnaire for Braley et al.’s ingroup/outgroup misperception study [1]. Cronbach’s α assesses how consistently a set of items measures the same underlying trait—higher values indicate that the model responses (approximating individual human users) are not random or contradictory but reflect a coherent underlying persona.
>
> This measure is directly related to the notion of stability. If a persona exhibits stable characteristics, we expect its responses across conceptually related items (e.g. ingroup and outgroup questionnaires) to align in a consistent direction. Also, Cronbach’s α is calculated by pooling within-person covariance across items, it provides evidence that our method does not merely match population-level statistics, but also produces internally coherent personas whose responses reflect stable psychological orientations—just as real humans do.
>
> In our results, the Cronbach’s α for our method is 0.649 (±0.025), closely aligning with the human benchmark of 0.664 (±0.037). The “±” values represent the 95% confidence intervals. The overlap between these intervals suggests no statistically significant difference, indicating that our virtual personas demonstrate a level of internal consistency comparable to that of real individuals.
>
> **Table R1. Cronbach’s alpha for Ingroup/Outgroup Misperception Questionnaires**
>
> |       | Cronbach’s α    |
> | ----- | --------------- |
> | Ours  | 0.649 (± 0.025) |
> | Human | 0.664 (± 0.037) |
>
>
> ### Qualitative Analysis of Plausibility and Stability of Backstories
> Additionally, we include a qualitative analysis of generated backstory samples: we have included two randomly sampled backstories (one matched to a Democratic user and another to a Republican) in our manuscript Appendix B, one of which reproduced as a truncated summary below. In the examples below, we observe that the continuations of backstory remain plausible across multiple turns of questions and responses: the author is a Indian American songwriter who resides in New Jersey. The rich details provided in the narrative—how the individual first studied English literature in college, enjoys cultural diversity around the neighborhood, and leans toward moderately liberal in political affiliation—is all expressed without notable incoherence, hinting at overall plausibility and stability in persona.
>
> We will include the analyses of internal consistency and qualitative discussion of the backstory samples in the final manuscript.

---

### Comment · Program_Chairs · 2025-04-03

This paper violates the page limit due to adding a limitation sections beyond the page limit. COLM does not have a special provision to allow for an additional page for the limitations section. However, due to this misunderstanding being widespread, the PCs decided to show leniency this year only. Reviewers and ACs are asked to ignore any limitation section content that is beyond the 9 page limit. Authors cannot refer reviewers to this content during the discussion period, and they are not to expect this content to be read.

---

> ### Author Response · Authors · 2025-06-01
>
> We appreciate the decision of leniency regarding the extra page provisioning for the limitation section and ethics statement. We will adjust the location and length of the sections appropriately in the final manuscript revision.

---

### Decision · Program_Chairs · 2025-07-08

**Decision:**

Accept

**Comment:**

The reviewers were unanimous in their recommendation to accept this paper (all 7s). Additionally, the authors provided very thorough responses to reviewer concerns, including new experiments on non-political domains (ATP Waves 34 & 99), detailed demographic breakdowns showing consistent improvements across subgroups, and quantitative stability analyses. These additions substantially strengthen the paper and I recommend acceptance.

As a bit of editorializing, I particularly appreciated the authors' response to 3XgY's concerns, which added both quantitative stability measures and qualitative examination of backstory samples. The qualitative analysis demonstrates plausible continuity across multi-turn interactions. Future work might further systematize such analyses to better understand what specific narrative features enable higher-order binding (beyond those that enable accuracy in individual self-opinions).